# The Influence of Surfactants, Dynamic and Thermal Factors on Liquid Convection after a Droplet Fall on Another Drop

**Sergey Y. Misyura \*, Vladimir S. Morozov and Oleg A. Gobyzov** 

Institute of Thermophysics Siberian Branch, Russian Academy of Sciences, Lavrentiev Ave. 1,
630090 Novosibirsk, Russia; morozov.vova.88@mail.ru (V.S.M.); oleg.a.g.post@gmail.com (O.A.G.)

\* Correspondence: misura@itp.nsc.ru; Tel./Fax: +7-383-3356577

**Abstract:** The regularities of the processes and characteristics of convection in a sessile drop on a hot wall after the second drop fall are investigated experimentally. The movement of a particle on a drop surface under the action of capillary force and liquid convection is considered. The particle motion is realized by a complex curvilinear trajectory. The fall of droplet with and without surfactant additives is considered. Estimates of the influence of the thermal factor (thermocapillary forces) and the dynamic factor (inertia forces) on convection are given. The scientific novelty of the work is the investigation of the simultaneous influence of several factors that is carried out for the first time. It is shown that in the presence of a temperature jump for the time of about 0.01–0.1 s thermocapillary convection leads to a 7–8 times increase in the mass transfer rate in drop. The relative influence of inertial forces is found to be no more than 5%. The fall of drops with surfactant additives (water + surfactant) reduces the velocity jump inside the sessile drop 2–4 times, compared with the water drop without surfactant. Thermocapillary convection leads to the formation of a stable vortex in the drop. The dynamic factor and surfactant additive lead to the vortex breakdown into many small vortices, which results in the suppression of convection. The obtained results are of great scientific and practical importance for heat transfer enhancement and for the control of heating and evaporation rates.

**Keywords:** sessile drop; drop impact; surfactant; convection

## 1. Introduction

### 1.1. Promising Technologies Based on the Interaction of Falling Liquid Drops and Heated Surfaces

Surface cooling by a spray (drop aerosol) is widely used in practice [1,2]. In this process, it is important to properly model the interrelated processes of heat and mass transfer, as this is what determines the efficiency of the technology as a whole. Rather frequently used are multicomponent solutions where convection in liquid depends on a set of key factors [3]. The continuous fall of drops of water–salt solution is realized on the tube walls in falling-film evaporators at desalination and cooling [4]. The dynamics of drop falling are important to consider in the chambers of internal combustion engines [5]. When the rate of the drop fall on a layer of burning fuel increases, three different modes are realized: spraying-injecting, splashing-injecting-secondary-injecting and bubble splashing [6]. The dynamics of the drop fall are important to consider when using fire suppression technologies [7,8]. The drop impact on the surface and the collective interaction of drops is important to consider in the following technologies: plasma spraying, inkjet printing, spray cooling power and electronic devices, and at increasing thermal comfort in the room and outside it [9].

The impact of single- and multicomponent liquid drops on a heated wall and liquid film was considered in [10,11].

*1.2. Modern Understanding of the Processes of Interaction of Falling Liquid Drops and Heated Surfaces*

Evaporation modes depend on the simultaneous influence of thermal and dynamic factors, which significantly complicates both experimental research and the construction of a theoretical model. The thermal aspect is largely related to the temperature of the wall, which is associated with four distinct evaporation regimes: (1) film evaporation, (2) nucleate boiling, (3) transition boiling, and (4) film boiling [12]. Dynamic factors lead to the regimes of the drop impingement, grouped into five different impact patterns: (1) completely wet; (2) wet film boiling; (3) transition; (4) dry rebound, and (5) satellite dry rebound, where dry impact implies that no liquid–solid contact occurs during the impact process. The number *We* has a stronger influence on dry wall impact than that on wet wall impact [12]. It has been found that when the impact time scale $t_d = d_{01}/U_{01}$ (where $d_{01}$ is the diameter of the falling drop and $U_{01}$ is the velocity of the falling drop before droplet interaction) is of the order of the thermal time scale $t_T = \lambda_w q_w c_w/h^2$ or larger (where $\lambda_w$ is the wall thermal conductivity, $c_w$ is the wall specific heat, and $h$ is the heat transfer coefficient), the effects of heat transfer (thermal factor) on impact behavior cannot be neglected, and the drop will inevitably contact the wall directly. If $t_T$ is longer than $t_d$, the wall remains isothermal, and the impact is not affected by the heat transfer [13,14]. In this case, only the dynamic factor is decisive. Different modes of high-temperature evaporation of droplets depending on the properties of the solution, wall temperature and wettability are considered in [15–17].

Surfactants affect not only the surface properties of the solution, but also the heat exchange in the drop, as the number of Marangoni and surface forces change. The peculiarities of solution wetting using surfactants are considered in [18–20]. Experimental data on free convection in a sessile evaporating drop show that theoretical predictions overestimate the experimental data on the water velocity dozens of times, which is associated with the influence of natural surfactants (contaminants in the form of surfactants) [3,21,22]. The use of surfactants reduces the surface tension of the liquid, which leads to a change in the modes of evaporation of the sessile droplet [23,24]. The evaporation kinetics of solutions with surfactant are similar to the droplet kinetic of pure aqueous solutions without surfactant. The main differences of surfactant solutions are as follows: (1) the lower values of initial contact angles, (2) the larger values of initial diameters of the droplet base, and (3) dependence of the receding contact angle on time for the second time stage at concentrations below the critical wetting concentration (CWC) [23]. For a drop of pure water at constant contact radius ($R_d$ = constant), the static contact angle depends only on the evaporation rate. The evaporation of the water droplet with surfactant results in an increase in surfactant concentration and its redistribution between the bulk and interfaces due to a decrease in droplet volume. As a result, the contact angle will decrease over time. Surfactants play an important role in suspensions. Surface textures enhance heat and mass transfer near a wall [25,26]. A small concentration of surfactants in emulsions allows reaching small droplet sizes and prevents the droplets' merger and the growth of their diameter [27,28]. The behavior of droplets on the heated wall was considered in [29,30].

*1.3. Modern Research Methods*

For correct modeling of heat and mass transfer and evaporation in the interaction of droplets, data on the instantaneous velocity fields inside droplets are required. In recent years, optical non-contact methods have been rapidly developing, and their resolution and accuracy have been increasing. The following methods are now widely used for the diagnostics of multiphase flows: Particle Image Velocimetry (PIV) [31–33], Particle Tracking Velocimetry (PTV) [34], and Laser-Induced Phosphorescence (LIP) [35]. The main features of specified methods application in drops and thin films are considered in [36–39].

Thus, the literature analysis has shown that most of the experimental studies of droplet falling on film or on the liquid layer are associated with the measurements of the following key parameters: the geometry of crown splashing (rim crown wall, cavity below film surface, crown diameter and height, crown shape, crown height evolution), ejecta sheet and multi-drop impact, and the rates of

droplet scatter at crown formation [40] with the use of PIV. In modeling the interaction of droplets, as a rule, the effects of buoyancy and Marangoni forces on convection generation inside the liquid are not taken into account. It is also unclear how the inertia forces affect the velocity field inside a droplet. Basically, the thermal and dynamic aspects concern the interaction of the droplet with the wall when the dynamic and thermal time ratio is considered. In the study of the properties of solutions with surfactants, the emphasis is placed on the study of wettability, contact angle dynamics and mapping the droplet evaporation modes. The effect of wettability with a high temperature is considered in [41]. High-temperature non-isothermal desorption was considered in [42]. The amalgamation and separation of droplets of different liquids leads to different flow regimes in mini- and microchannels [43–45]. A change in component concentrations during the amalgamation leads to free convection in the liquid.

*1.4. Research Objectives*

An analysis of the literature has shown that previous research was aimed at studying the drop shape behavior after the drop fell on a solid wall. The fall of drops on the liquid layer led to the formation of crowns and splashes. There are practically no experimental data on the effect of an instantaneous local temperature jump on the free surface of a drop on the convection inside the drop. The question remains how the interaction of droplets and the indicated short-term temperature inhomogeneity affect the intensity and duration of convection, as well as what characteristic convective structures occur inside the drop.

The first objective is to identify the roles of the dynamic factor (inertia forces) and the thermal factor (thermocapillary forces) in the generation of convection in the liquid when the small drop falls on the large sessile drop, located on a hot wall.

The second objective is to study the effect of surfactants on heat and mass transfer in the drop. In this case, the role of the droplet shape, contact angle and evaporation on convection inside the droplet is not considered due to the short-term interaction of two droplets. This formulation of the problem allows for clearly identifying the factors that control convection, as well as answering the question of how the surfactant affects the instantaneous velocity field after the drop falls on a thin layer of liquid.

## 2. Experimental Setup and Procedure

The scheme of the experimental setup for measuring the instantaneous velocity field inside drop 2 and the temperature field $T_s$ on the surface of the sessile drop is shown in Figure 1.

During the experiment, the external pressure of 1 bar, relative humidity of 35–36%, and temperature of the external air of 21–22 °C were constant in the specified ranges throughout the experiment. Drop 2 with initial volume $V_{02} = 40$ μL (initial diameter of drop 2 was $D_{02} = 7$–7.5 mm, and the initial height $h_{02} = 2.6$ mm) was placed on a horizontal heated wall. Drops 1 and 2 were applied by dispensers, positioned perpendicular to the wall surface (Figure 1) (using a single-channel electronic pipette Finnpipette Novus (Thermo Fisher, Vantaa/Joensuu, Finland) with a step of the volume change of 0.1 μL). All the experiments were repeated four times, and the ranges of $h_{02}$ and $D_{02}$ are given taking into account repeated measurements of the geometric parameters of droplet 1.

Droplet 1 ($V_{01} = 2.5$ μL, $d_{01} = 0.8$ mm) fell on drop 2 (Figure 1) ($V_{02} = 40$ μL). The height of the droplet 1 fall ($H$) measured from the edge of the dispenser to the top of the large drop 2 was 4–5 mm. The Weber number was $We = (\rho U_{01}^2 d_{01})/\sigma = 1$–1.3, where $\rho$ is the density of water and $\sigma$ is the surface tension coefficient for water-air. The rate of droplet 1 falling just before the contact with the top surface of the sessile drop 2 was $U_{01} = 0.32$ m/s. This velocity was determined by high-speed shooting. Neither drop broke up, and at an impact on the liquid surface there was no spray.

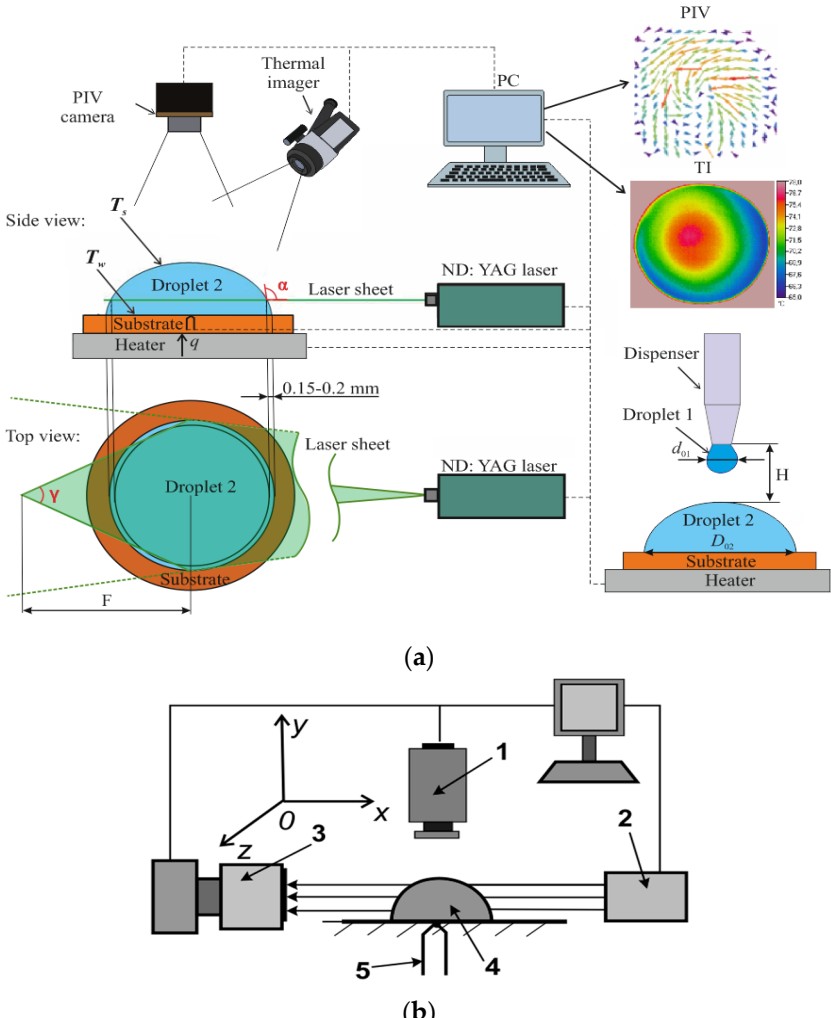

**Figure 1.** (**a**) Scheme of measuring $T_s$ and the instantaneous velocity field inside drop 2 using PIV. PIV—Particle Image Velocimetry; TI—Thermal Imager. (**b**) The measurement scheme of drop 2 static contact angle: (1) the video camera; (2) the source of plane-parallel light; (3) the camera Nikon D750 (with micro lens); (4) drop 2; (5) a thermocouple for measuring the wall temperature ($T_w$).

Droplet 1's impact did not lead to the formation of the crown, which was confirmed by the measurements by the high-speed camera. After droplet 1's fall, the diameter of the base of large drop 2 $D_{02}$ did not change. After the interaction of droplets, a thermal wake remained on the free surface of drop 2. The substrate was made of copper (the substrate thickness was 5 mm and its diameter was 50 mm). After each experiment, the substrate was treated several times with alcohol and water, and then thoroughly dried. In addition, the wall roughness was periodically monitored by a profilometer, which showed the invariance of the mean square value of wall roughness. The substrate was placed on a hard table, located on a base that excluded vibrations of the table and the substrate. Squeezing out a large sessile drop was implemented rather slowly to maintain the constancy of the drop geometry for different experiments. The drop diameter and height for repeated experiments differed by no more than 3%.

The static contact angle ($\theta_0$) for the sessile water drop 2 was 87–90° at a wall temperature under the sessile drop $T_w$ = 79–81 °C. The optical system of plane-parallel light generation was used to provide a shadow image of drop 2 (Figure 1b). To obtain such light, the source (MI-150) and the telecentric backlight illuminator (62-760, Edmund Optics) were used together with the glass fiber optics cable (BX4 type Dolan-Jenner) and video camera (FastVideo 500 M) with the macro lens Sigma 105 mm f/2.8 G IF-ED AF-S. The initial static contact angles were determined using a tangential method.

To minimize the measurement error of the contact angle, the experiments were repeated three times (under the same conditions). The average value of the contact angle was determined by the results of three experiments. The maximum measurement error of the initial static contact angle of the sessile drop did not exceed 3–5%.

To measure the temperature field on the surface of the sessile droplet, a thermal imager with multiple image magnification was applied. The interfacial temperature of drop 2 was determined with the help of thermal imager NEC R500EX-Pro (NEC Avio Infrared Technologies, Yokohama, Japan) (spectral range of 8–14 μm, frame frequency of 30 frame/s, measurement accuracy of ±1 °C, and thermogram resolution of 640 × 480). The spectral bandwidth of infrared camera in short wave range (SW) was 3–5 μm.

The substrate with drop 2 was located on the surface of the heating plate (the plate was heated by electric current and tungsten wire). The wall temperature $T_w$ was adjusted automatically with an accuracy of ±1 °C. To measure the temperature $T_w$, a low-inertia platinum-rhodium thermocouple with inertia of 0.1 s (the junction diameter of 0.05 mm) was used. The relative error of $T_w$ temperature measurement did not exceed ±0.5 °C. This measurement error was within the adjustment accuracy $T_w$ (±1 °C). The thermocouple was fixed on the substrate surface by means of thermal paste with high thermal conductivity. The thermocouple practically did not overhang the wall surface and did not distort the velocity field in drop 2.

In different experiments, the drop 2 consisted only of the distillate, and the composition of the droplet 1 changed: (1) 100% water; (2) water + surfactant "AF 9-12" (NEONOL AF 9-12 oxyethylated monoalkyl phenol) with mass concentration of 4%; (3) water + surfactant "OP-10" (auxiliary material OP-10 product of treatment of a mixture of mono-and dialkylphenols with ethylene oxide) with mass concentration of 1%; (4) water + surfactant "Sodium DS" (sodium dodecyl sulfate) (0.1% mass). For all aqueous solutions, surfactant concentrations exceeded the critical concentration at which micelles are formed. These concentrations were taken to avoid a study on the effects of concentrations in this article due to the large amount of experimental data. The effect of concentrations will be presented in the next work.

For measuring the velocity field in a horizontal section of the droplet, the Particle Image Velocimetry method was used. All measurements using PIV were carried out only in the horizontal section of the sessile drop (Figure 1). The horizontal measurement section of drop 2 was at a distance of 0.15–0.2 mm from the substrate surface. To measure the instantaneous velocity fields, a double solid-state Nd:YAG laser Quantel EverGreen 70 was used, which had the following main parameters: wavelength—532 nm, repetition frequency—4 Hz, and pulse energy—70 mJ. For the formation of the laser sheet, cylindrical lenses with an opening angle of 22° were used. For the purposes of the laser sheet positioning, an optical mirror was used. Registering images of drops required the camera ImperX IGV-B2020M with basic settings: image resolution—2048 × 2048 pix, frequency of shooting—4 fps, and bit width—8 bit. Nikon macro lens (200 mm f/4 AF-D Macro) was also used. To process experimental data on the velocity field inside drop 2, the software Actual Flow with software packages PIV was used. The plane of the laser sheet was parallel to the wall and was at a height of 0.15–0.2 mm from the wall surface. Since the contact angle of the drop was close to the right angle, the value of the angle α corresponded to 85–87°. As a result, the error of the drop's curvature practically did not influence the results of measuring the instantaneous velocity field. The maximum error in measuring the average velocity of the liquid in a given horizontal section of the drop, taking into account the above-mentioned measurement features, did not exceed 15–20%.

When processing experimental data on instantaneous velocity fields inside drop 2, three types of velocity were determined:

(1)　$U_{\mathrm{max}}$—absolute maximum value;

(2)　$U_{\mathrm{max}(20)}$—maximum value resulting from averaging over 20 maximum values;

(3)　$U_{\mathrm{aver}}$—average absolute value.

## 3. Results and Discussion

### 3.1. Velocity Field in Sessile Water Drop, Located on a Heated Wall

Drop 2 evaporated in the mode of constant contact radius (CCR). The fall of the second small droplet 1 did not lead to a change in the radius $R_{02}$ of sessile drop 2 ($R_{02}$ = const). Figure 2a presents experimental data for velocities in the water drop 2 (initial volume of drop 2 ($V_{02}$) is 40 μL; and the wall temperature $T_w$ = 80 °C).

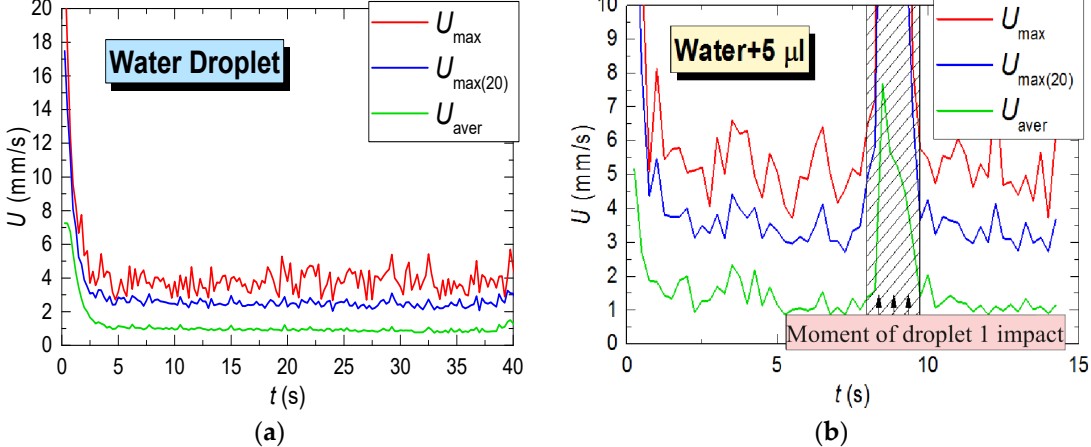

**Figure 2.** (**a**) Change of characteristic velocities $U_{max}$, $U_{max(20)}$, and $U_{aver}$ in the horizontal section of the water drop 2 ($V_{02}$ = 40 μL; $T_w$ = 80 °C); (**b**) change of characteristic velocities $U_{max}$, $U_{max(20)}$, and $U_{aver}$ in the horizontal section of sessile drop 2 at droplet 1 falling ($V_{02}$ = 40 μL; $V_{01}$ = 2.5 μL; $T_W$ = 80 °C).

The velocities $U_{max}$, $U_{max(20)}$, and $U_{aver}$ were measured by PIV. The time $t$ = 0 corresponds to that when drop 2 spread along the hot wall, and the radius of the base of drop 2 $R_{02}$ was established constant. The experimental data in Figure 2a were obtained without the small droplet 1 falling. The maximum value of the average velocity $U_{aver}$ is 7.2–7.5 mm/s and corresponds to the time close to $t$ = 0.2–0.5 s (the initial time of placing the cold drop 2 with a temperature of 20 °C on a hot wall). The maximum value $U_{aver}$ was recorded at the initial moment, since this time corresponded to the maximum temperature difference ($\Delta T_s$) on the free surface of the drop ($\Delta T_s = T_w - T_0 = 80 - 20 = 60$ °C, where $T_w$ is the wall temperature, and $T_0$ is the initial temperature of droplet 1). The thermal Marangoni number (Equation (1)) is:

$$Ma_T = (\Delta T_s h/\mu a)\cdot(d\sigma/dT_s), \tag{1}$$

where $\Delta T_s$ is the temperature gradient on the liquid, $\sigma$ is the surface tension coefficient for water-air, $h$ is the drop 2 height ($h = h_{02}$ = 2.6 mm), $\mu$ is the dynamic viscosity of water, and $a$ is the thermal diffusivity of water. The Rayleigh number (Equation (2)) is:

$$Ra = g\beta\Delta T_s(h)^3/va, \tag{2}$$

where $v$ is the kinematic viscosity of the water, $\beta$ is the coefficient of thermal expansion, and $g$ is the gravity acceleration. Indeed, the equation of motion is nonlinear. The convection introduces nonlinearity. However, within the limits of the experimental error it is possible to carry out a qualitative assessment in a linear approximation. Let us consider the total convective velocity as a sum of individual components. Let us make an approximate estimate for $U_{aver}$ [32,33,46] between the average velocity in drop 2 and the Marangoni and Rayleigh numbers (Equation (3)):

$$U_{aver} = U_{MT} + U_{Ra} = k_T(Ma_T + Ra) \tag{3}$$

where the empirical constant $k_T$ determined from experimental data is $0.15 \cdot 10^{-7}$ (m/s).

This linear approximation is quite justified, since the Rayleigh number is much less than the *Ma* number, and the nonlinear velocity term must be significantly less than the total value of $U_{\text{aver}}$.

Quasi-linearity also follows from quasi-stationarity and linear superposition of forces. This does not take into account the shape of the drop, the contact angle, or the number of vortices in the drop. However, it is currently impossible to accurately model the convection field inside the drop, since the noticeable suppression of free convection due to surfactant on the liquid surface is not taken into account. The influence of surfactant leads to a tens of times underestimation of the theoretical value of the free convection velocity in the water drop, compared to the experiment [21,22]. A decrease in the convection velocity is also observed at low concentrations of alcohol [3]. The empirical constant $k_T$ is obtained by generalizing experimental data and indirectly takes into account the influence of surfactant, which allows applying Equation (3) to approximate the value of the average convection velocity not only in the drop, but also in a thin layer of a single-component liquid. For the case of solution, an additional term associated with the action the solutal Marangoni number appears.

The velocity value drops rapidly and already in the first 3–5 s enters the quasi-stationary level. Such a rapid decay would be impossible if the nonlinear member was comparable with the average total value. This velocity field would be extremely unstable, which does not correspond to the experimental data. However, it is impossible to estimate experimentally the nonlinearity because the total error in determining the calibration coefficient $k_T$ will exceed 10–15%. In addition, nonlinearity can play a significant role in the theoretical solution of the differential equation. The considered approach applies an estimate based on experimental data, rather than the theoretical solution. In this case, the $U_{MT}$ value includes the effect of *Ra*, since the buoyancy in drop 2 is always present along with the thermocapillary convection. Thus, the $k_T$ coefficient and the $U_{\text{aver}}$ include nonlinearity. Since $k_T$ is determined at a significant temperature difference, this nonlinearity is close to the maximum, and the decrease of $\Delta T_s$ will only lead to a decrease in the role of nonlinearity.

Taking into account the linear approximation (Equation (3)), we obtain the value $U_{\text{aver}} = k_T(Ma_T + Ra) = 0.15 \cdot 10^{-7}(410{,}000 + 57{,}000) = 0.007$ m/s = 7 mm/s (to calculate the $Ma_T$ and *Ra* numbers (Equations (1) and (2)) we use the following values: $h = 2.6$ mm, $\Delta T_s = 60$ °C, $d\sigma/dT = 0.17 \cdot 10^{-3}$ (Nm$^{-1}$K$^{-1}$), $a = 16.3 \cdot 10^{-8}$ (m$^2 \cdot$s$^{-1}$), $\mu = 0.4 \cdot 10^{-3}$ (Pa·s), $\beta = 0.35 \cdot 10^{-3}$ (K$^{-1}$). The experimental value of maximum $U_{aver}$ is 7.2–7.5 mm/s (Figure 2b), which closely corresponds to the calculated value (7 mm/s). Some discrepancy may be due to inaccuracy of measurement of drop 2 height and the error of determining the coefficient in the formula for Marangoni. In addition, the velocity value in a particular section is not exactly equal to the average volume velocity. Therefore, the velocity will be slightly different in different sections. It is also difficult to determine with high accuracy the average temperature difference $\Delta T_s$ for the initial time moment ($t = 0$–2 s), since during this time period the temperature distribution $T_s$ on the drop surface is highly uneven. In addition, the wall temperature $T_w$ under drop 2 also has a highly unsteady character for $t = 0$–2 s.

As you can see from Figure 2a, the velocities $U_{\text{max}}$, $U_{\text{max}(20)}$, and $U_{\text{aver}}$ are greatly reduced during the first 5 s because of the liquid heating. After positioning the drop on the heated wall during the first 3–5 s, the temperature of the drop surface $Ts$ increases from temperature of the external air of 20 °C to 52–55 °C (Figure 3). Further, the temperature of drop 2 surface $T_s$ varies moderately, i.e., $T_s$ values continuously increase, and the difference $\Delta T_s$ (Figure 4) on the contrary, continuously decreases. These changes are slow. Therefore, the problem can be considered for $t > 5$–10 s as quasi-isothermal and quasi-stationary.

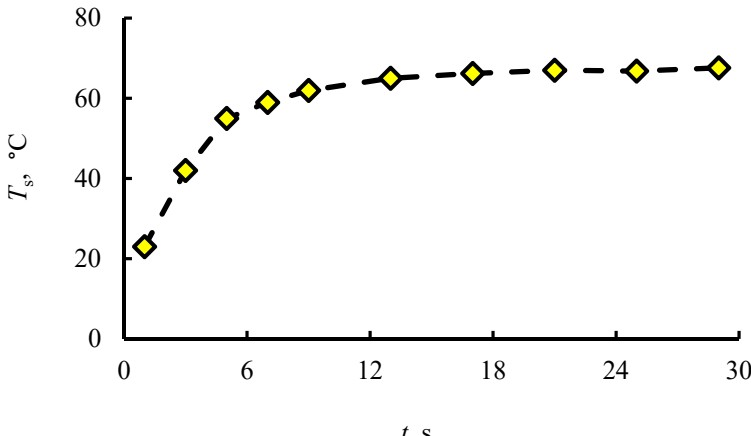

**Figure 3.** Changes of average temperature for the entire surface of drop 2 over time.

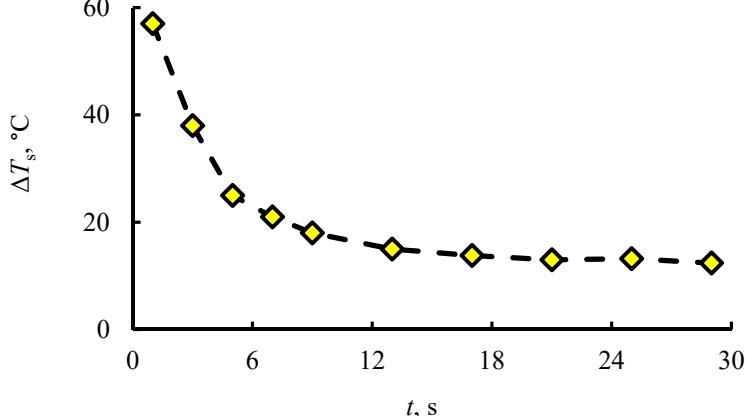

**Figure 4.** Changes in average temperature difference $\Delta T_s$ for the entire surface of drop 2 over time.

### 3.2. Velocity and Temperature Field for Sessile Water Drop after the Fall of Another Droplet of Water

Figure 2b shows experimental data for the case when the small cold droplet 1 of water falls vertically on the sessile drop 2 of water located on a heated wall. The fall of droplets in all experiments occurs approximately in 6–7 s after placing the large drop 2 on the heated wall. For the time $t = 6$–7 s, the velocity is approximately constant, and the temperature difference before falling between the wall and the interface is approximately $\Delta T_s = 12$–14 °C. As the graph shows, the interaction of the two drops has led to a sharp jump of the average velocity up to 7.2 mm/s. After about 2.5 s after the small droplet 1 falls, $U_{aver}$ takes a value corresponding to the velocity before the fall, i.e., about 1–1.2 mm/s. Usually, estimates in calculations and experiments are given for the maximum velocity. From Figure 2b it can be seen that the maximum velocity $U_{max}$ is several times higher than the average $U_{aver}$. The value of the maximum velocity strongly depends on the method of its determination, i.e., there can be no clearly defined method for calculating the extremum due to significant unsteadiness of the process and random behavior for the instantaneous velocity vector. In these studies, the main focus is on $U_{aver}$, since this velocity is used for qualitative and quantitative approximations (a relationship is established for free convection velocity, buoyancy and thermocapillary forces). However, the empirical coefficient has been obtained earlier for the average velocity in the drop $U_{aver}$.

To describe the Marangoni flow, it is important to know the change in the drop surface temperature $T_s$ and the temperature difference $\Delta T_s$ with time $t$. Figure 3 shows experimental data on the change in the average temperature for the entire drop surface, which was determined over the entire thermal field at the time under consideration. The average temperature was determined by ten different circles drawn inside the drop interface. This algorithm corresponded to the software. The circles were drawn at a certain distance from the contact line of the drop to exclude the influence of the wall.



The average temperature was determined for each circle. The average surface temperature of the drop was calculated as the average value for ten circles. The error of averaging using this method was less than the measuring error of the thermal imager. The graph of changes in temperature $T_s$ consists of two modes: rapid growth of $T_s$ over time during the first 5–7 s after placing the drop on the hot wall, and a slow increase in the temperature of the free liquid surface for $t > 7$ s (quasi-stationary thermal mode). The fall of the cold small droplet 1 on the sessile hot drop 2 is realized at the very beginning of the quasi-stationary mode. Based on experimental data for the $T_s$ field, a graph of the change in $\Delta T_s$ over time is constructed (Figure 4). There the temperature difference $\Delta T_s = T_w - T_s$ is defined as the average value of the temperature difference for the entire surface of drop 2 (the $T_s$ values are taken from the graph in Figure 3). It is obvious that the modes of change of $\Delta T_s$ coincide with Figure 3. Thus, based on the data in Figure 4, the maximum values of thermocapillary forces and buoyancy forces will correspond to the first few seconds after placing drop 2 on the wall. The excess of $\Delta T_s$ almost three times for the initial period, in comparison with the quasi-stationary thermal regime, leads to a three-fold increase in the convection velocity for the initial period of time in contrast to $t > 7$ s.

Figure 5a gives the temperature field of the surface of drop 2 using the thermal imager. On the basis of the thermal images (Figure 5a), the hottest area of drop 2 surface is in its center, and when moving towards the contact line $T_s$ decreases (except for a very narrow area near the contact line).

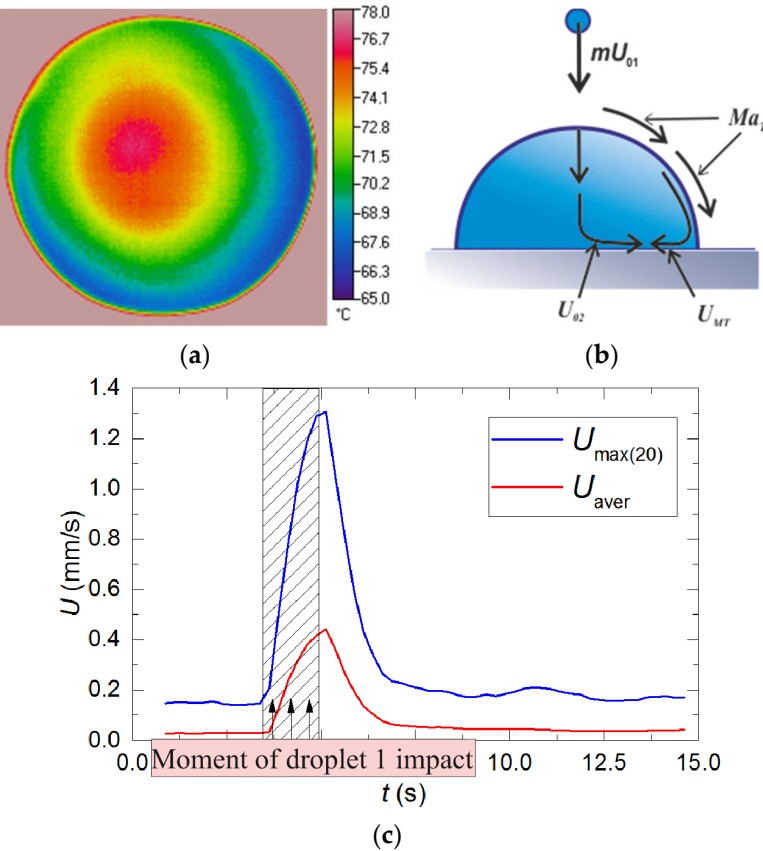

**Figure 5.** (**a**) Thermal image of water drop 2 interface ($Ma_T$—the thermal Marangoni number, $t = 5$ s); (**b**) the direction of characteristic velocities for the two limiting cases of interaction of drops (the joint influence of dynamic and thermal factors); (**c**) velocities in the horizontal section of sessile drop 2 ($V_{02} = 40$ μL; $V_{01} = 2.5$ μL; $T_w = 20$ °C).

The direction of the temperature gradient cannot be related to the measurement error of the thermal imager, i.e., to the effect of the wall. The drop 2 height of 2–3 mm is much higher than the lowest possible height, when there is an effect of the wall on the infrared radiation for the water layer [47] (the water layer thickness should be no less than 0.3–0.5 mm). Even for the specified small

height of the liquid, the effect will also be negligible. The direction of rotation of the liquid in the drop and the value of the average velocity depend on the ratio of forces: buoyancy ($Ra$), surface force due to the temperature gradient on the free surface of the drop ($Ma_T$) and the friction force. The direction of rotation will be determined by the value and the direction of $Ra$ and $Ma_T$. Then, the thermal Marangoni flow ($Ma_T$) will be directed from the center of drop 2 to the contact line (towards the greater surface tension of water $\sigma$) (Figure 5a).

It is important to note that in this work the direction of the temperature gradient on the interface and of the $Ma$ do not coincide with the generally accepted ones, i.e., from edge to center. The flow direction towards the center of drop 2 occurs with weak convection, i.e., when the conductive heat transfer plays a determining role. In this case, the coldest point on the surface of the sessile drop 2 will correspond to its center. Then the maximum surface tension will also take place in the central part of drop 2 surface. In the present work, convection significantly exceeds the conductive transfer and determines the direction of the liquid rotation. At the initial moment of stretching drop 2 on the surface of the hot wall, buoyancy determines a certain direction of rotation, which afterwards remains. This direction of circulation is not caused by the error of the thermal imager, as indicated above.

Figure 5b shows a diagram of velocity directions for the following options: $U_{02}$ and $U_{MT}$. The pulse from the falling droplet 1 ($mU_{01}$) is completely transmitted to the sessile drop 2 on a hot wall.

Let us consider the factors that can lead to free convection in the sessile heated drop after the small cold drop falls. After falling, a pressure jump occurs in the bottom surface of the sessile drop [48]. The duration of the pressure surge is microseconds. In the present work, measurements of the pressure inside the drop were not performed. Further, a transition process is implemented, when an excess of pressure leads to convection inside the drop and to a change in its shape, as well as to fluctuations in the free surface of the sessile drop. These factors are called dynamic. Since the pressure jump and interaction of elastic waves inside the droplet last for microseconds, the action of viscosity forces can be neglected during this time. Changes in the shape of the large drop when two drops interact can be neglected as well, since the contact line of the drop remains stationary, and the volume of the small falling drop is much smaller than the volume of the sessile drop. One can also neglect changes in temperature due to deformation of the free surface of the drop caused by fluctuations. One dynamic factor remains: the change in the static pressure field and the organization of liquid rotation in the sessile drop 2 after droplet 1 falls. To date, there are no analytical and numerical solutions that would allow modeling all these factors. In addition, one can also add another factor: the organization of many vortices inside the drop due to the instability of free convection inside the drop 2. In connection with the above, experimental data on convection, as well as simple estimates that would allow assessing the role of key factors in the transfer of heat and momentum in the sessile drop, are of interest. It should be noted that this work is not theoretical and does not aim to obtain strict estimates. The basic idea of estimates is quite simple. As a result of the interaction of two drops, convective motion is realized due to the dynamic factor (discussed above) and the thermal factor: the appearance of a short-term local temperature difference on the surface of the large drop. If simplified simulation of convection due to a temperature jump in accordance with Equation (3) corresponds satisfactorily to the experimental data, then we can assume that the dynamic factor has a negligible influence at a set falling height of droplet 1 and the volume of the falling drop.

Figure 5c provides characteristic velocities for the specified isothermal case ($T_{02} = T_{01} = T_w = 20\,°C$). As can be seen from the graph, after the fall of the small droplet 1, in large drop 2 there was a jump of $U_{aver}$ ($U_D$) only by 0.4 mm/s.

The direction of the Marangoni force ($Ma_T$) in Figure 5b is shown before the droplet 1 impact. After the droplet impact, the direction of the $Ma_T$ will change to the opposite direction (the coldest place of $T_s$ will correspond to the center of drop 2). Then, the direction velocities from the dynamic and thermal factors will be the same. Thus, the total influence of thermal and dynamic factors leads to the velocity $U_{aver} = U_{MT} + U_D = 7 + 0.4 = 7.4$ mm/s. The obtained calculated value coincides with the experimental one (7.2–7.5 mm/s in Figure 2b). At that, the role of the dynamic factor is only 5–6%.

Since the predominant effect is exerted by the thermal factor, it is important to consider the features of the thermal field change on the surface of sessile drop 2 immediately after the fall of the second droplet 1. For this purpose, the thermal imaging measurements were carried out. It was important to determine how quickly the temperature gradient on the surface of the large drop 2 falls with time (after the fall of droplet 1). The maximum gradient $\Delta T_s = 60\ °C$ should appear immediately after the interaction of drops.

Figure 6a shows the velocity fields (PIV measurement) at the interaction of two drops in isothermal conditions ($t = 0$ s corresponds to droplet 1 falling time). The non-circular horizontal section of drop 2 (Figure 6a,b) is caused by the appearance of "blind spots" due to the focusing of the laser sheet by the drop. Such zones were "disguised" in order to avoid the appearance of erroneous vectors in the final velocity field. As a result, the velocity field is asymmetric. The cross-correlation algorithm for calculating the velocity used in this work implies splitting the image into elementary regions ($64 \times 64$ pix regions were used) and finding the maximum of the correlation function in each region. At the same time, small local peaks of luminosity appeared periodically in the image of drop 2 due to the reflection of the laser from a large group of tracers. There were local peaks of velocity. When analyzing the results of experiments, such peaks were not taken into consideration. Figure 6b shows images of drop 2 before and after lightening the image in the image editor. In Figure 6b, the outline of drop 2 is clearly visible (highlighted with a red line).

Figure 7 presents the magnified thermal pictures of the drop 2 interface after the fall of droplet 1. At time $t = 0$ s there is a symmetrical temperature field (droplet 1 did not contact the surface of drop 2, droplet 1's temperature was 20 °C, and for drop 2 $T_s = 67$–70 °C. The purpose of thermal imaging measurements is to show the time of thermal relaxation ($t_T$) after the interaction of cold and hot drops. Already in $t = 0.05$ s after the merging of drops, the temperature on the free surface becomes quasi-stationary, i.e., the time of thermal relaxation is less than 0.1 s. From the contact line of the drop to the center of the drop, the distance on the free surface is approximately equal to $l_S = 5.5$ mm/s. You can estimate the velocity of the liquid on a free surface as $Us = l_S/t_T = 5.5$ (mm/s)/0.05 (s) = 110 mm/s. The maximum value for the average velocity in drop 2 ($U_{aver}$, Figure 2b) is 7.2 mm/s. Thus, $Us$ exceeds $U_{aver}$ approximately 16 times. This strong velocity suppression is due to the vortex formation inside the droplet, as well as due to the viscosity. Estimating how many times the velocity on the liquid surface ($Us$) at the moment of the thermal jump exceeds the thermal front velocity ($U_a$) due to the molecular temperature conductivity $a$. $U_a = a/l_S = 0.03$ mm/s. Then, $Us/U_a = 3700$. These estimates demonstrate that the surface thermocapillary forces play a large role in comparison with molecular transport when the thermal boundary conditions on the free surface of the liquid change sharply. An abrupt local change in surface tension due to the temperature $T_s$ or due to the surfactant will result in a sharp velocity jump ($Us$), which will quickly attenuate in an extremely short period of time.

This conclusion is extremely important for the correct modeling of a large number of fast-flowing phase transition processes when there is a phase boundary (liquid–gas). One of them is the plasma spraying of the nanosurface. Temperature inhomogeneities on the surface of the melt during crystallization will lead to high velocities on the surface of the melt (liquid). In this case, to estimate this velocity, it is necessary to take into account the thickness of the liquid layer for calculating the friction force.

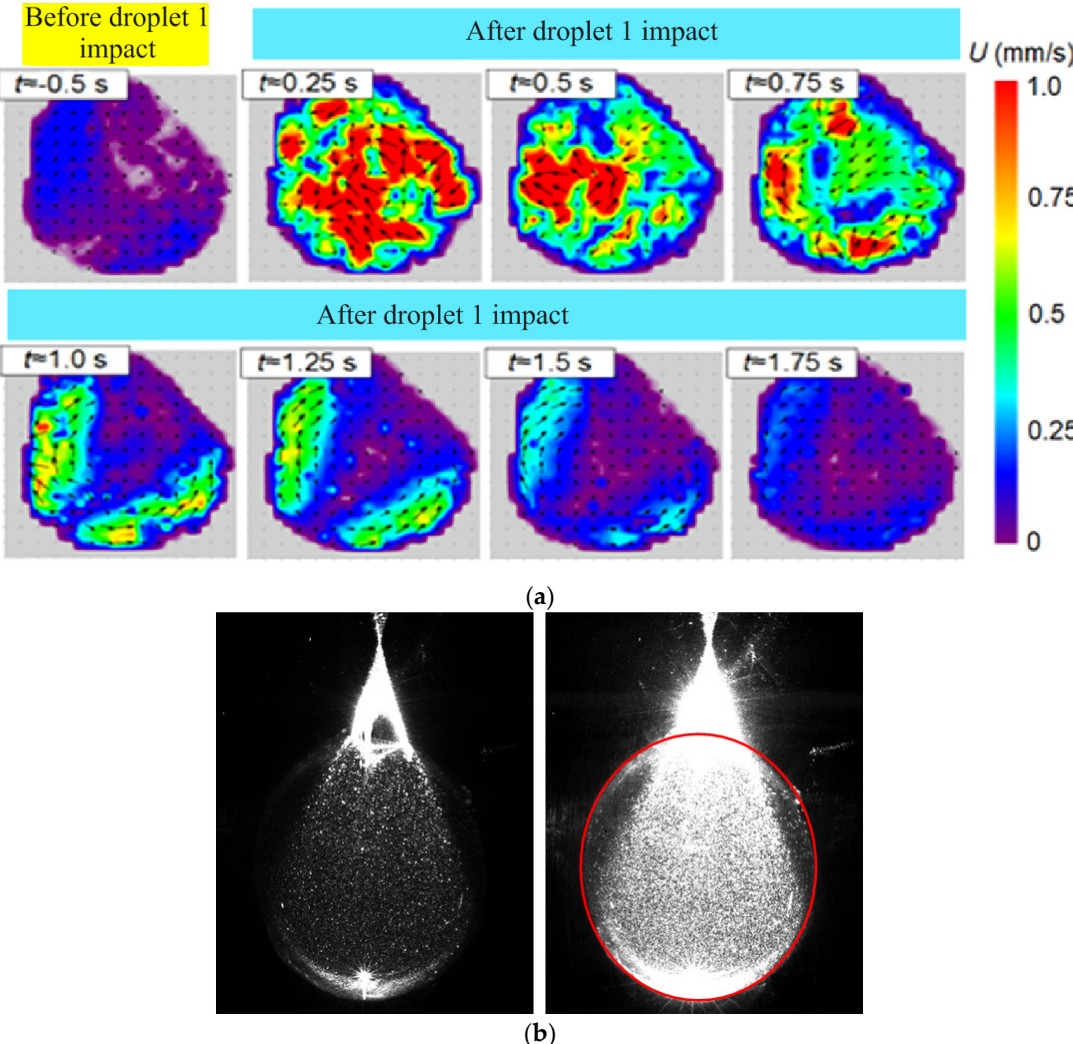

(a)

(b)

**Figure 6.** (**a**) Instantaneous velocity fields at the interaction of two drops (PIV measurement, starting point $t = 0$ s corresponds to droplet 1 falling time, $V_{02} = 40$ μL; $V_{01} = 2.5$ μL; $T_w = 20$ °C); (**b**) image of drop 2 before (left) and after (right) image lightening in the image editor (the red line indicates the contour of drop 2).

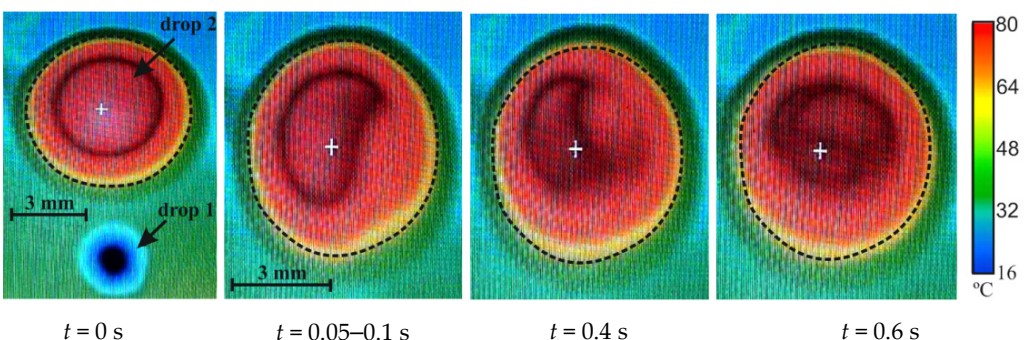

**Figure 7.** Thermal imager measurements of thermal field on the surface of drop 2 (water) after the fall of droplet 1 (water).

As it is seen from Figure 7, already after 0.05–0.1 s additional cooling of drop 2, the surface will be only by 3–5 °C lower than for $t = 0$ s. The temperature difference on drop 2's surface in 0.05 s will be $\Delta T_s = 14 + 4 = 18$ °C, rather than 60 °C. According to Equation (3), an increase of $\Delta T_s$ by

20–30% will result in a value of $U_{aver}$ = 1.6 mm/s. This value is about 5 times lower than the maximum experimental value for $U_{aver}$ = 7.2–7.5 mm/s. Thus, a short-term temperature gradient for the interval $\Delta t$ = 0.01 s has led to a four-fold increase in the velocity due to thermocapillary convection. Figure 7 also shows a dotted line for the drop. After this line, the thickness of the water layer becomes less than 0.3–0.5 mm and the $T_s$ measurement is incorrect due to the influence of the substrate on the temperature measurement, i.e., the $T_s$ values are underestimated.

### 3.3. The Effect of Surfactants on the Velocity Field Inside Sessile Drop after an Impact of Another Small Droplet

Theoretical predictions [3,49–51] for the free convection velocity inside sessile drop 2 of water or an aqueous solution with a low concentration of another liquid show that the numerical calculation overestimates the velocity value tens of times compared to the experiment [3,52]. Multiple velocity reduction in the experiment is associated with the presence of contaminants in water, which cannot be eliminated [51,53]. The exception is an aqueous solution of alcohol with a high concentration of alcohol [3,50]. The suppression of a coffee-ring occurs at a certain form of micron-sized particles as a result of capillary forces [54]. It is assumed that the impurities reduce the surface tension gradient $\sigma$ on drop 2 surface $d\sigma/dl$, where $l$ is the distance on the free surface of drop 2. Thus, even without adding the surfactant to the liquid, there is a significant impact of natural surfactants.

There are many technical problems when the use of surfactants contributes to both the growth and reduction of the reaction rate. For example, the addition of low-concentration methanol to oil contributes to the control of hydrate plugs in pipelines [55–57] by lowering the hydrate formation temperature. Surfactants in the form of SDS (sodium dodecyl sulfate) significantly increase the rate of hydrate formation. The formation of micelles from surfactant molecules can increase the rate of hydrate formation 700 times [58] since micelles are active centers of hydrate formation. A small concentration of surfactants in the liquid suspension in droplet-gas suspension allows achieving small droplet sizes and prevents the merger of droplets and the growth of their diameters.

To date, there is very little experimental data on the effect of surfactants on the velocity field inside drop 2. Most of the works are devoted to the study of physical and chemical properties and wettability, as well as the evaporation mode when adding surfactants [23,24]. It is extremely difficult to study the hydrodynamics of two colliding droplets in a moving gas flow using PIV. This process can be studied by the example of falling of small droplet with surfactant additives on the sessile drop, located on a hot wall.

Figure 8a–c present experimental data on the effect of surfactants of the following types: AF 9-12; 4% mass; OP-10; 1% mass; and Sodium DS; 0.1% mass. The falling droplet 1 consisted of a water + surfactant solution. Sessile drop 2 consisted of distillate. The surface $Ma_T$ flow was directed from the center of drop 2 to its edge (Figure 5b,c), since $\sigma$ increases in the direction of the edge. In 2.0–2.5 s after droplet 1 falling, the average velocity in the section of drop 2 enters the quasi-stationary mode, i.e., the velocities before and after the drop are equal. The surfaces of AF 9-12 and OP-10 did not change the contact radius of drop 2 (the contact line of drop 2 did not shift). The surfaces of SDS led to a jump in the contact line.

An arrow in Figure 8c indicates the time at which the diameter of the contact line increased abruptly (approximately for 1 s) by 10–20%. The use of all three types of surfactants has shown a significant decrease in the average velocity jump. The use of AF 9-12 and OP-10 has led to about 1.8–1.9 times decrease in $U_{aver(max)}$ (for pure water $U_{aver(max)}$ = 7.5 mm/s (Figure 2b), and for AF 9-12 and OP-10 $U_{aver(max)}$ = 4.0 mm/s (Figure 8a,b). The use of SDS has led to the maximum suppression of the convection velocity (the maximum jump of the average velocity $U_{aver(max)}$ is 2.5 mm/s), i.e., $U_{aver(max)}$ decreases about 3 times. A stronger decrease of $\sigma$ for water + SDS (Figure 8c) has led to a non-equilibrium of the contact line of drop 2.

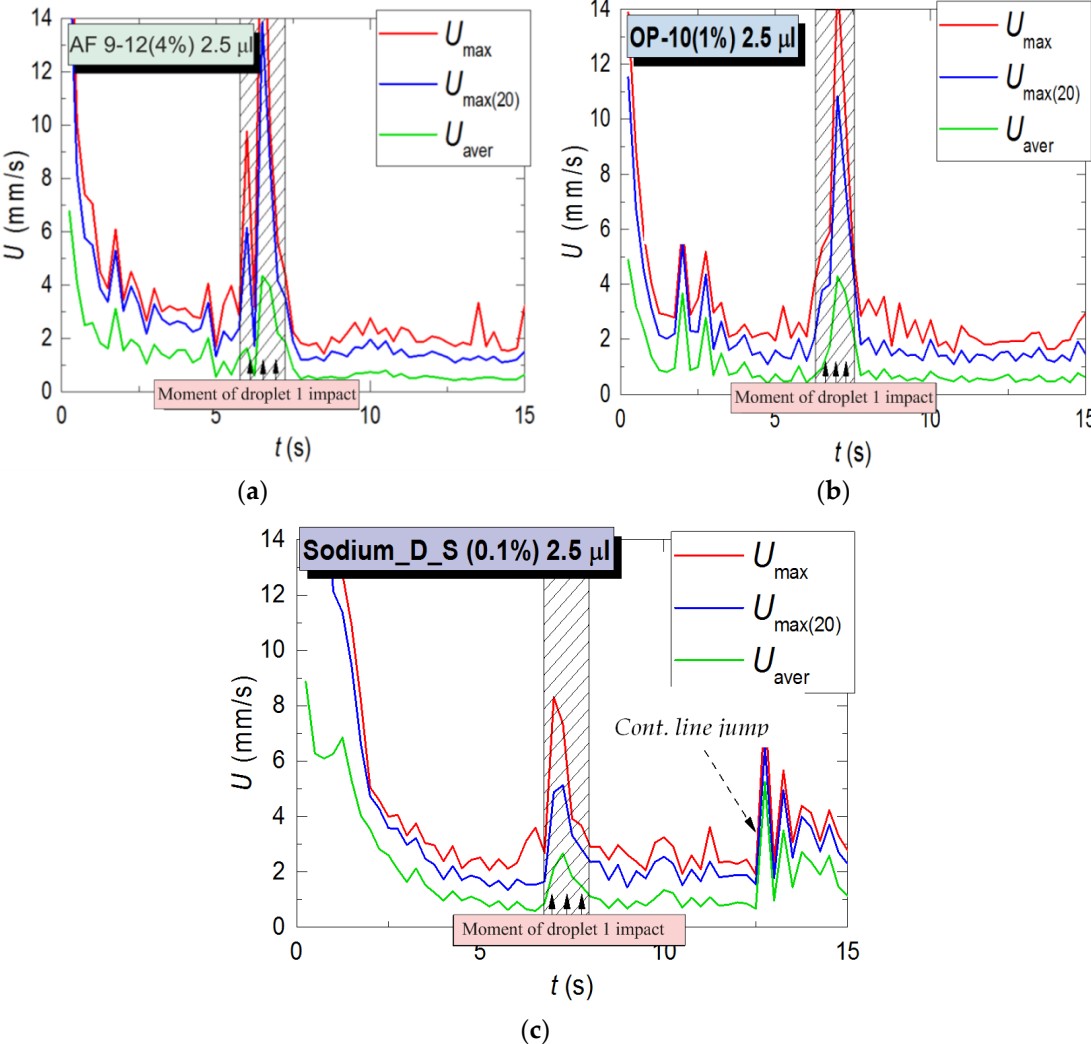

**Figure 8.** (**a**) The behavior of $U_{max}$, $U_{max(20)}$, and $U_{aver}$ at falling of droplet 1 ($V_{01}$ = 2.5 μL; water + surfactant AF 9-12; 4% mass) on sessile drop 2 ($V_{02}$ = 40 μL; $T_w$ = 80 °C; water); (**b**) the behavior of $U_{max}$, $U_{max(20)}$, and $U_{aver}$ at falling of droplet 1 ($V_{01}$ = 2.5 μL; water + surfactant OP-10; 1% mass) on sessile drop 2 ($V_{02}$ = 40 μL; $T_w$ = 80 °C; water); (**c**) the behavior of $U_{max}$, $U_{max(20)}$, and $U_{aver}$ at falling of droplet 1 ($V_{01}$ = 2.5 μL; water + surfactant Sodium DS; 0.1% mass) on sessile drop 2 ($V_{02}$ = 40 μL; $T_w$ = 80 °C; water).

### 3.4. Instantaneous Velocity Fields in a Horizontal Section of Sessile Drop

It is quite difficult to measure the velocity field directly at the moment of the fall (for example, at $t$ = 0.01–0.1 s) because of the extremely fast change in the direction of the vectors in time and because of the rapid change in the velocity in time. In Figure 9, time $t$ = 0 s corresponds to the time of drops' contact. Accordingly, when $t$ is less than zero, there is no drop impact, and for $t > 0$, the image after the interaction of drops is given.

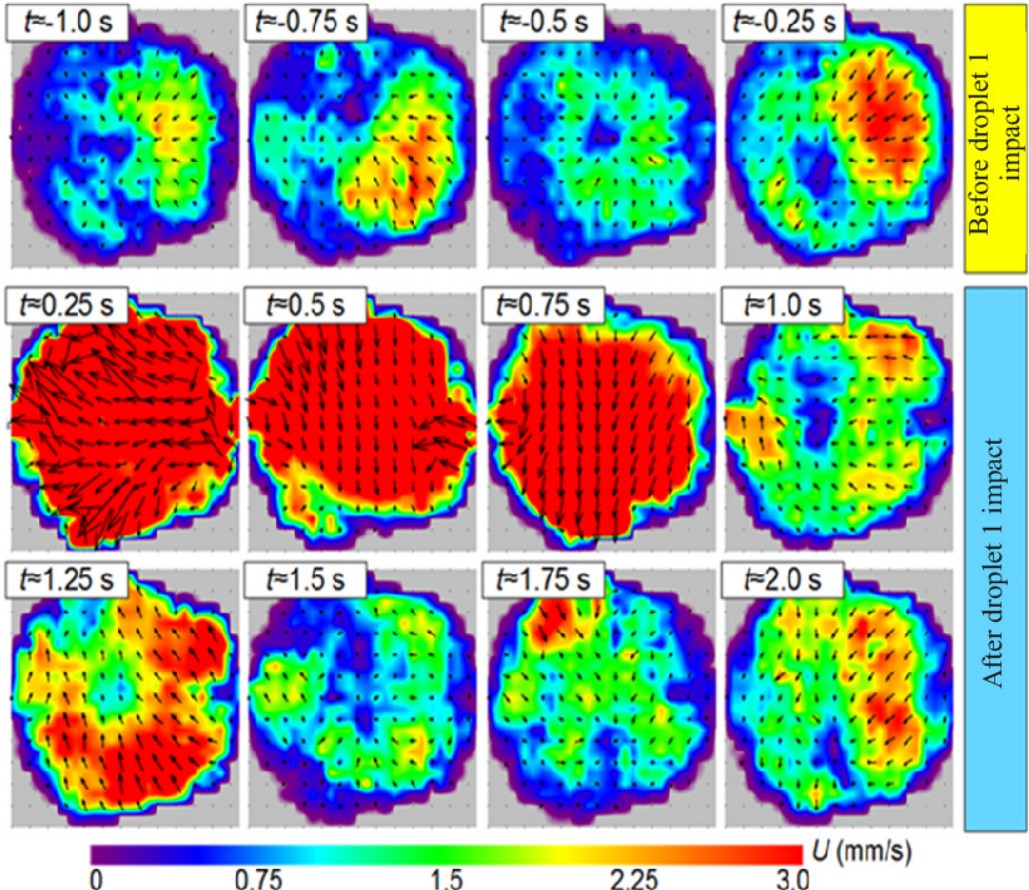

**Figure 9.** Velocity field in horizontal section of drop 2; $V_{01} = 2.5$ μL; $V_{02} = 40$ μL; $T_w = 80$ °C; $T_{01} = 20$ °C; water (droplet 1) + water (drop 2).

After the fall of the droplet, on the surface a predominant direction of velocity appears, resulting in a single main vortex ($t = 0.75$ s), the size and energy of which substantially exceed those of other smaller vortices ($t = 0.25$–$0.5$ s). In the vicinity of the wall ($t = 0.75$ s), the velocity vectors (in the horizontal plane) have the same direction (parallel to each other). In this case, the diameter of the vortex coincides with the diameter of the drop. It is obvious that near the free surface of the drop, the direction of the liquid rotation will change to the opposite. This flow pattern is fundamentally different from Figure 6a, when even a very small value of *We* leads to unstable rotation of the liquid and to fragmentation of the vortices. Already in 0.25 s, several smaller vortices are formed. It is obvious that a significant increase in the value of *We* will lead to an increase in the number of small vortices. Thus, thermocapillary forces (Figure 9) lead, on the contrary, to stable rotation. This conclusion is very important, as the Marangoni forces will lead to the maximum convection velocity. The thermal factor has a much greater effect (compared to the dynamic factor) due to the stability of the surface flow in the presence of thermocapillary forces.

It is important to emphasize some limitations related to stability. The aqueous solution of alcohol leads to instability of the surface flow and to chaotic behavior [3], i.e., in this case, surfactant in the form of alcohol forms an uneven field of concentrations on the drop surface. Below we will focus on this property, which is very important for correct modeling.

In Figures 10–12, sessile drop 2 consisted of water and the falling droplet 1 consisted of an aqueous solution (water + surfactant). As can be seen from Figures 10–12, the velocity field, after the droplet fall, becomes uneven. Against the background of the main direction of motion there are individual vortexes that change their position and direction of rotation. The absence of a symmetrical toroid in the drop and the presence of many small vortices, as well as a rapidly changing pattern of the velocity

field in the drop section is probably due to the nonuniform distribution of surfactant concentrations on the drop 2 surface, which leads to different surface tension gradients $d\sigma/dl$ (uneven distribution of $\sigma$ on the free surface will result in gradients $\Delta\sigma/\Delta l$).

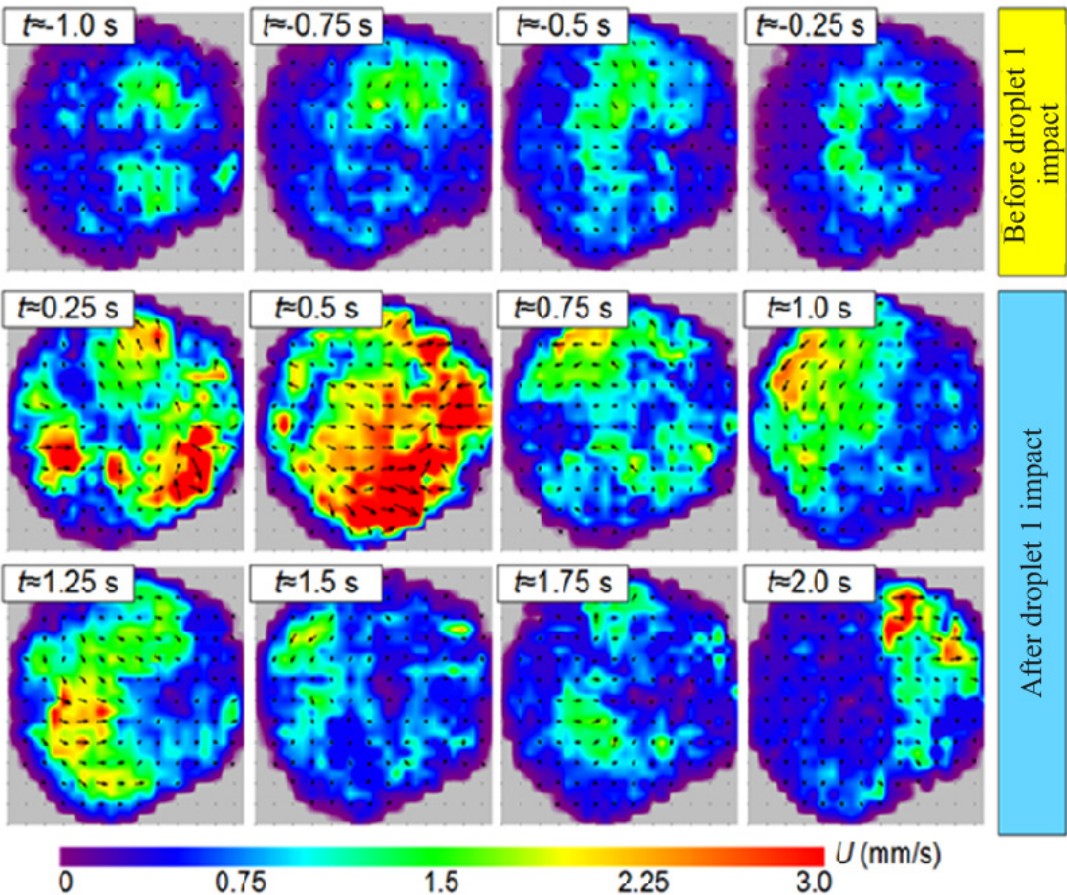

**Figure 10.** Velocity field in horizontal section of drop 2; $V_{01} = 2.5$ μL; $V_{02} = 40$ μL; $T_w = 80$ °C; $T_{01} = 20$ °C; water (drop 2), water + surfactant OP-10) (droplet 1).

For approximately 2 s after the droplet fall, the average velocity in the section continuously decreases. After $t = 1.75$ s (Figure 11) the velocity increases slightly since the amplitude of the average value is commensurate with the random pulsation value. The flow inside drop 2 is unstable, and the vortex is continuously redistributed from one area to another.

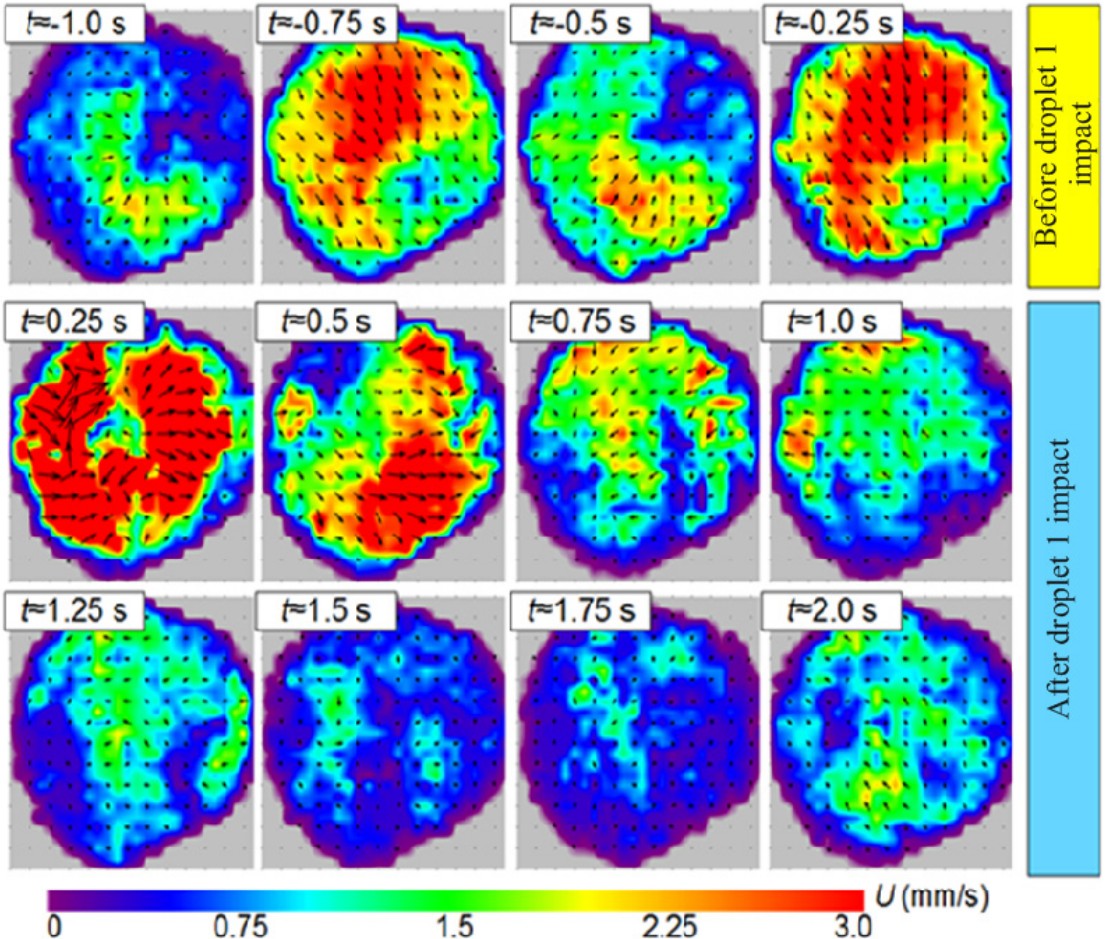

**Figure 11.** Velocity field in horizontal section of drop 2; $V_{01}$ = 2.5 µL; $V_{02}$ = 40 µL; $T_w$ = 80 °C; $T_{01}$ = 20 °C; water (drop 2), water + surfactant AF 9-12 (NEONOL AF 9-12: oxyethylated monoalkyl phenol) (droplet 1).

Let us estimate the time for the spread of surfactant molecules over the surface of sessile drop 2. The diffusion coefficient $D_s$ on drop 2 surface for water–SDS is assumed to be the same as in the volume ~ $10^{-10}$ [59]. The characteristic diffusion time on drop 2 surface (diffusion time of relaxation) $t_D = l^2/D_s = (5.5 \cdot 10^{-3} \text{m})^2/10^{-10} = 30 \cdot 10^4$ s ($l$—distance on drop 2 surface). Based on the estimates of the diffusion time, it turns out that the rate of spread of surfactant molecules on drop 2 surface is determined not by diffusion, but by the surface flow of Marangoni, which causes convection inside drop 2. Then, the convective relaxation time $t_{c(s)}$ when using surfactant will be $t_{c(s)} = l/U_{aver} = 5.5$ mm/1.5 mm/s = 2.8 s (where the average velocity $U_{aver}$ (Figure 8c) for the time interval $\Delta t$ = 4–5 s corresponds to 1.4–1.5 mm/s. Thus, already in 2 s after the fall of droplet of water with surfactant, a surface layer of surfactant should be formed near the contact line of the sessile drop 2. The contact line should move in 2–3 s, and in reality, the increase in the contact radius occurs approximately 7 s after the fall, i.e., with a noticeable delay (the characteristic time $t_m$ is 2–2.5 times more than $t_{c(s)}$).

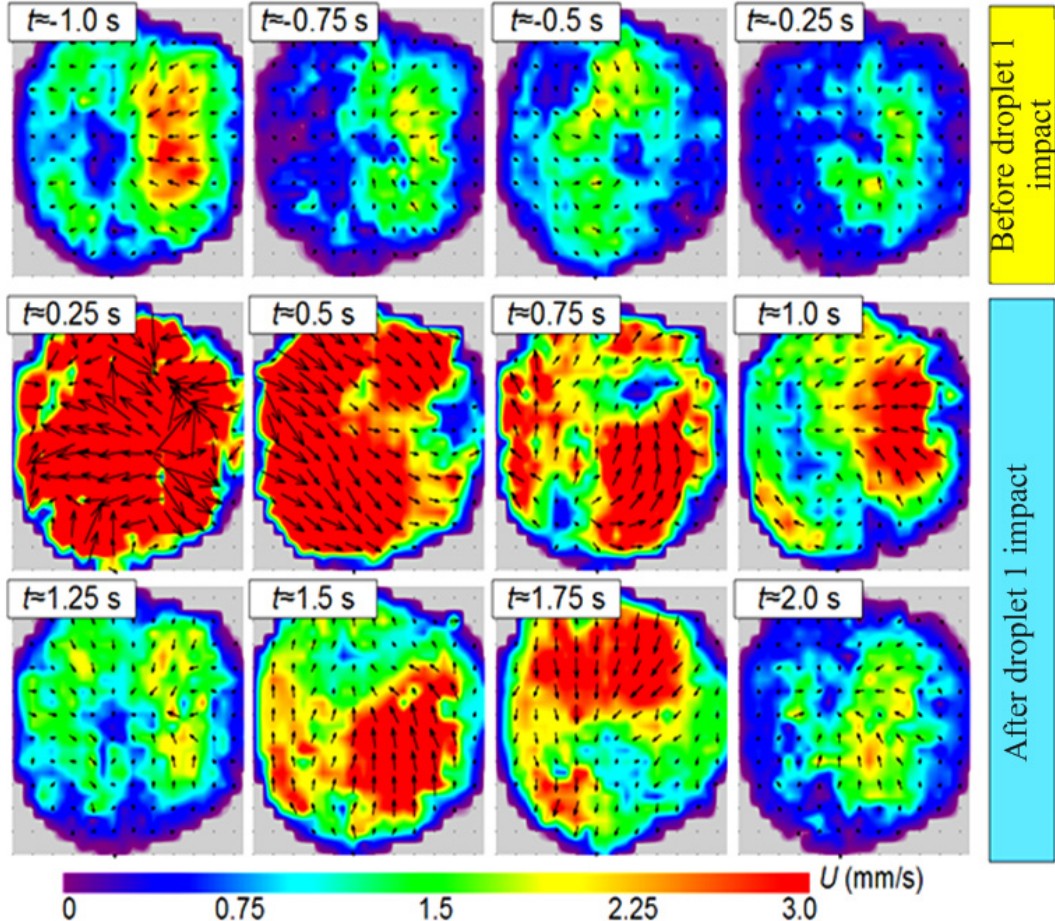

**Figure 12.** Velocity field in horizontal section of drop 2; $V_{01}$ = 2.5 µL; $V_{02}$ = 40 µL; $T_w$ = 80 °C; $T_{01}$ = 20 °C; water (drop 2), water + surfactant SDS (sodium dodecyl sulfate) (droplet 1).

To explain such a noticeable delay, an additional experiment was conducted. Several metal particles with a diameter $d$ = 0.1–0.2 mm were placed on a surface of a water drop after its positioning on a hot wall ($T_w$ = 80 °C). The surface Marangoni flow and the capillary forces led to the displacement of the particles that were monitored using video cameras at multiple magnification of the image. Figure 13a shows the trajectory of the particle in time.

In the presence of a toroid inside the drop, the motion of the particle must be realized along the trajectory 1 (red line). The real trajectory of motion according to experimental measurements corresponds to trajectory 2 (blue line). The direction of the velocity vector has a general tendency towards the center of the toroid. As noted earlier, the Marangoni flow is directed from the center of the toroid to the edge of the drop. Thus, the particle motion due to capillary forces is impeded by Marangoni forces and the friction force of the liquid. Therefore, the capillary force exceeds the sum of Marangoni forces and friction.

For points of trajectory 2, the velocity values are given in mm/s. When the particle moves from the drop edge to the center and along the quasi-straight line, the acceleration occurs, and the particle reaches the maximum velocity (2.9 mm/s). As a result of the turn, the velocity decreases almost three times. When moving in a direction parallel to the contact line, the velocity is almost constant (1.2 mm/s). When approaching the center of the toroid, the velocity is reduced to a minimum value (less than 0.5 mm/s). The average particle velocity for the entire trajectory is 1.5–2.0 mm/s, which closely corresponds to the velocity value in the drop cross section $U_{ever}$ = 1–1.2 mm/s (Figure 2, $t > 5$ s). The velocity on the drop surface according to numerical calculations) is about 2 times higher than the velocity in the horizontal section, which is close to the wall. The results of the

trajectory measurement have shown an amazing result. The real trajectory is curved and has a random character. Indeed, there is a general tendency of particle displacement to the center of the drop. However, the particle motion is superimposed with random oscillations, which are caused by three-dimensional vortices inside the drop. These vortices move from one place to another, change the direction of rotation and interact with each other. As a result of such chaotic trajectory, the length of the entire trajectory from the contact line of the drop to its center will be 2–3 times longer than for the shortest line on the curved surface. As a result, the time $t_m = 3t_{c(s)} = 6$ s, which corresponds well to the experiment.

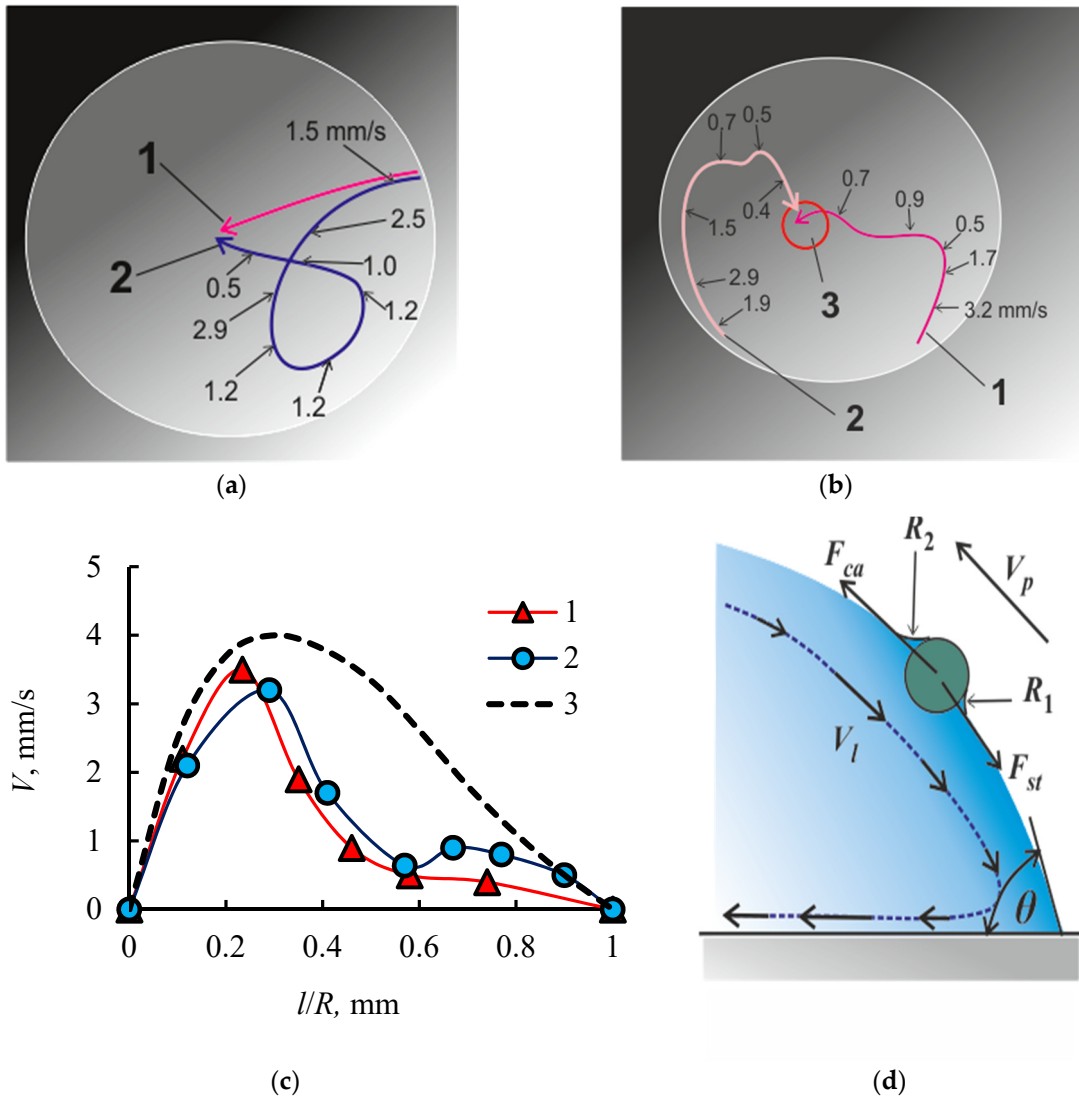

**Figure 13.** (**a**,**b**) Trajectory of the particle on the drop surface: 1, 2—particle trajectory; 3—the area of the toroid center; (**c**) particle velocity on the drop surface: 1, 2—experiment; 3—modeling by (4); (**d**) forces acting on the particle.

Figure 13b illustrates the trajectories of the other two particles. The initial position of the particles was in the vicinity of the contact line. The particles moved to region 3, where the toroid has its center. In region 3, the liquid velocity is close to zero, since the flow lines turn around, and the velocity changes direction by 90° (the velocity is directed inside the drop). Figure 13c provides experimental data for the particle velocity on the drop surface (curves 1 and 2). The distance l corresponds to the drop surface ($l/R = 0$ refers to the contact line of the drop, $l/R = 1$ for the center of the toroid, where the liquid velocity

is zero, $l$ is the projection on a straight line (for example, on line 1 in Figure 13a), i.e., $l$ is not the length of the particle trajectory).

The movement of the salt crystal near the contact line under the action of capillary force is considered in [60]. To date, there is only an empirical expression describing the movement of a micron-sized particle on the surface of the drop [61].

Let us consider the equation of motion of a particle in a simple model approximation, taking into account basic forces applied to a particle (Figure 13d (the Stokes force ($F_{st}$) and the capillary force ($F_{ca}$)). The acceleration of a particle is determined by the equality of forces in accordance with (4):

$$m dV_p/dt = F_{ca} - F_{st} = 2\sigma S_1((R_2 - R_1)/R_2 R_1) - 6\pi r \mu (V_p - V_l), \tag{4}$$

where $m$ is a particle mass; $V_p$ is the velocity of a particle; $\sigma$ is the water surface tension; $R_1$ and $R_2$ are the radii of curvature of the meniscus; $S_1$ is the area of a particle, corresponding to the meniscus; $r$ is the radius of the $Al_2O_3$ particle ($r = 0.1$ mm); $\mu$ is dynamic viscosity; and $V_l$ is the fluid velocity at the drop surface. The boundary conditions of Equation (4) are at $l = 0$ (the location on the contact line) $\Delta R = R_1 - R_2 = \max$, $V_l = 0$; at $l/l_{max} = 1$ (the center of the toroid), the difference $\Delta R = R_1 - R_2 = 0$, $V_l = 0$. The liquid velocity from zero value reaches the maximum ($V_l = 3.5$ mm/s in accordance with the experimental data) and then tends to zero when approaching the center of the toroid). The curve of velocity change $V_l$ (from zero to maximum value of 3.5 mm/s and from maximum to zero value) was set as a cosine dependence, which closely corresponds to the known theoretical calculations for the velocity on the surface of the drop [21,22]. Estimates show that the thermocapillary force is several orders of magnitude greater than buoyancy and gravity due to the smallness of the particle size. Thus, to describe the particle motion on the free liquid surface, it is sufficient to consider the movement due to curvature, changes in $\sigma$ over the surface, and also due to the Marangoni flow (taking into account the change in $\Delta T_s$ over the surface of drop 2). Changes in $\Delta T_s$ were taken from the experimental data of the thermal imager, and taking into account that on the edges of drop 2 the temperature of the free surface of the liquid is approximately equal to a predetermined wall temperature ($T_w$).

In approximate calculations, it is assumed that the value of $\Delta R$ varies depending on the cosine. The surface flow of the liquid, in accordance with the experimental data, is directed against a particle motion, i.e., from the center of drop towards the drop contact line ($V_l < 0$). Curve (3) in Figure 13c is calculated according to Equation (4). Theoretical calculation was performed along the trajectory of the particle 1 (Figure 13a), i.e., on the shortest distance on the drop surface from the edge of the contact line to the center of drop. Since it was extremely difficult to experimentally measure the difference $\Delta R$ due to its low value, it was necessary to base the calculation on such a value of the curvature radii difference that corresponded to the maximum particle velocity of 3–4 mm/s. The simulation shows that the agreement with the experiment is achieved when the relative difference between the values of the radii $\Delta R_1 = (R_2 - R_1)/R_1$ is only a fraction of a percent.

Thus, the solution is very sensitive to changes in wettability and small surfactant additives can lead to a noticeable change in the particle velocity.

As it is seen in Figure 13c the theoretical curve reflects the qualitative behavior of the experimental curve. The maximum value of the calculated velocity is close to the experimental values. The segment on the abscissa axis $l/R$, corresponding to the velocity extremum and plotted from the coordinate origin, is approximately $l/R = 0.3$, the derivative $dV_p/dt = \max$ at $l/R = 0$, since $\Delta R = \max$; the derivative $dV_p/dt = 0$ at $l/R = 1$, since $\Delta R = 0$. The difference between experiment and simulation for the section of $0.3 < l/R < 0.8$ is quite obvious. The real trajectory of the drop is very different from a straight line. At the time of the trajectory half turn, the drop velocity falls several times. The specified interval has several points of turning. As a result, the velocity in this interval will be significantly less. To describe the trajectory rotation, it is necessary to take into account the three-dimensional unsteady flow inside the drop. The toroid is imposed by random circular movement. The vortices in the water drop change position and direction, thereby changing the direction of the particle.

The proposed calculation model also describes the experimental data of [61], in which the velocity of a particle with a diameter of 20 μm at a distance from the contact line of 250 μm is approximately $(2–3)\cdot10^{-4}$ mm/s. The calculation by Equation (4), taking into account the area $S_1$ for the specified diameter (20 μm) and neglecting the Marangoni forces, also gives the maximum velocity value of $(2–3)\cdot10^{-4}$ mm/s. In fact, the Marangoni flow without such particles develops the liquid velocity on the drop surface of 0.001–0.01 mm/s. This velocity exceeds by 10–100 times that of capillary motion. Obviously, the particles have suppressed the Marangoni flow in [61].

Figure 14 illustrates experimental data for the average velocity $U_{aver}$ in horizontal section of drop 2 ($V_{02} = 40$ μL; $T_w = 80$ °C).

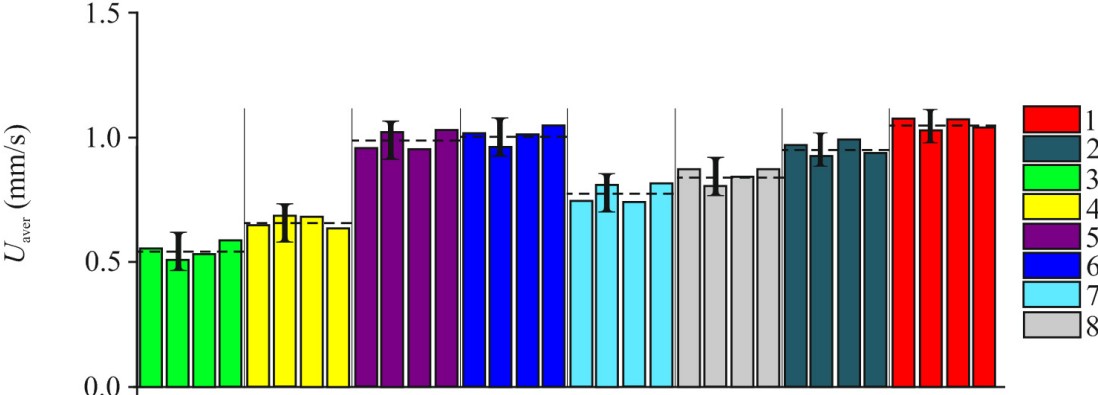

**Figure 14.** Velocity $U_{aver}$ in the horizontal section of sessile drop 2 ($V_{02} = 40$ μL; $T_w = 80$ °C); (1 and 2)—without the fall of small droplet 1; (3–8)—in 3 s after the fall of small droplet 1 with temperature of 20 °C (sessile drop 2 consists of water). Composition of drops: 1—sessile drop 2 (water); 2—sessile drop 2 (water with graphite particles); 3—droplet 1 (surfactant AF 9-12 (4%), $V_{01} = 5$ μL); 4—droplet 1 (surfactant AF 9-12 (4%), $V_{01} = 2.5$ μL); 5—droplet 1 (surfactant OP-10 (1%), $V_{01} = 2.5$ μL); 6—droplet 1 (surfactant OP-10 (1%), $V_{01} = 5$ μL); 7—droplet 1 (surfactant SDS (0.1%), $V_{01} = 5$ μL); 8—droplet 1 (surfactant SDS (0.1%), $V_{01} = 2.5$ μL); I is the interval of measurement errors and I is the interval of measurement errors relative to the dotted horizontal line (average velocity value over four repeated experiments).

For each specific type of mixture, four repeated experiments have been carried out. The time of measuring $U_{aver}$ in 3 s after droplet 1 falls corresponds to the quasi-stationary regime, i.e., when $U_{aver}$ does not change over time. Previous experiments have investigated the behavior of $U_{aver}$ immediately after the fall of droplet 1 (within the first two seconds after the fall). The purpose of these experiments is to study the influence of dynamic and thermal factors. Three seconds after the drops' interaction, only the surfactant effect remains on the surface of the sessile drop 2. Experiments with drops of different compositions in Figure 14 are compared with drop 2, which consists of pure water (1). Adding of graphite particles has led to insignificant decrease in $U_{aver}$ (2). The strongest effect of suppression of the average convection velocity is specific when applying surfactant AF 9-12 (3 and 4 in Figure 14). Surfactant SDS also decreases free convection in drop 2 (7 and 8 in Figure 14). The use of surfactant OP-10 had almost no action on $U_{aver}$ (5 and 6 in Figure 14). Falling droplet 1 (water + surfactant) passed the surfactant molecules to sessile drop 2. As a result of the fall, $d\sigma/dT$ decreased with respective decrease in the rate of convection inside drop 2.

It should be added that the performed experiments complement the previously developed ideas about the influence of a group of factors (thermal, dynamic, geometric) on the convection velocity in drops at different heating schemes (in particular, in [62] the conditions of convective heating were considered, and this work studies the conductive heating, but taking into account the fall of the second droplet 1). The research on the influence of heating conditions is deemed to be promising, for example, heat transfer (convective, conductive, radiative, and mixed) on the scales of the impact of these factors.

## 4. Conclusions

Experimental studies on the interaction of the falling droplet 1 and sessile drop 2 on a hot wall have been conducted. The temperature of the falling droplet 1 was 20 °C and the wall temperature was 80 °C.

The simultaneous influence of several key factors, dynamic and thermal, as well as the influence of surfactants, has been considered for the first time. The dynamic factor was related to the inertial forces and the pressure jump inside drop 2. The heat factor was associated with an increase in the surface temperature gradient after droplet 1 fall.

The novelty of the work is that a large influence of local temperature and concentration nonuniformity on the generation of intense convection inside the droplet is demonstrated. It has been shown that in the moment of interaction of drops there was a 7–8 times growth of velocity $U_{aver}$ inside drop 2, even with a very low velocity of droplet 1 falling (0.32 m/s) and when the Weber number $We = 1.3$. At that the influence of the dynamic factor was negligible. The use of SDS has led to a four-fold decrease in the maximum of the average velocity inside sessile drop 2.

Experimental studies of the instantaneous velocity fields inside sessile drop 2 have been conducted using Particle Image Velocimetry. The instantaneous velocity field changes significantly after the droplet 1 falls. Due to the use of surfactants, larger vortices become unstable and break up into smaller ones, which randomly rotate inside drop 2. The use of graphite particles of micron size has led to a decrease in drop 2 heating period and to a slight increase in the velocity at the initial moment of interaction of drops (velocity increase by 10–20%).

Measurement of the velocity field for a quasi-stationary regime, where the velocity in drop 2 has practically ceased to change with time (i.e., for a time greater than that of dynamic relaxation after the fall of small droplet 1), has shown that a significant velocity suppression (approximately two times) is observed only when using the surfactant (AF 9-12).

The studies have important practical applications. To intensify heat and mass transfer in a sessile or suspended drop, local heating or cooling of the free surface can be used. To suppress convection in a liquid during the interaction of two drops, a surfactant can be used for one of the drops. Thus, depending on the specific task, it is possible to both intensify and suppress heat and mass transfer in a drop.

**Author Contributions:** Investigation, V.S.M., O.A.G.; writing—review and editing, S.Y.M. All authors have read and agreed to the published version of the manuscript.

**Funding:** Research was supported by Russian Science Foundation (project 19-79-30075).

**Acknowledgments:** Russian Science Foundation (project 19-79-30075).

**Conflicts of Interest:** The authors declare no conflict of interest.

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
