# Peer review of "The Influence of Surfactants, Dynamic and Thermal Factors on Liquid Convection after a Droplet Fall on Another Drop"

_applsci, doi:10.3390/app10124414_

Round 1
Reviewer 1 Report
The paper has been thoroughly rebuilt since my previous review. All my previous doubts/questions have been adequately addressed by the authors. Moreover, the new content has significantly improved the scientific quality of the manuscript and it can attract more attention from the readers. Therefore I recommend to accept it in the previous form.
Author Response
Response to Reviewer 1 Comments
Point 1: English language and style are fine/minor spell check required. The paper has been thoroughly rebuilt since my previous review. All my previous doubts/questions have been adequately addressed by the authors. Moreover, the new content has significantly improved the scientific quality of the manuscript and it can attract more attention from the readers. Therefore I recommend to accept it in the previous form.
Response 1: The English language has been corrected.

Reviewer 2 Report
The paper presented an experimental study of a droplet impact on a sessile droplet previously placed on a hot wall. The temperature and velocity fields were measured by thermal imaging camera and PIV setups, respectively. The effect of surfactant on impact dynamics was studied as well. The manuscript has some novelty. The paper is well organized and written. I suggest publication after minor revisions.
(1) Line 192 of pg 6, why did the authors choose to average over 20 maximum values? It seems that 20 is an arbitrary number.
(2)How is constant kT (line 226 of pg 6) determined? If it is determined from authors' own data, it is no wonder they can get a good match of velocity predicted from the eq. 3 with experimental values.
(3)How is average temperature difference presented in fig. 4 calculated?
(4) I suggest the authors include images of dynamics process of droplet impact on sessile droplet, which can help better understand the effect of Weber number on velocity field measured by PIV.
Author Response
Response to Reviewer 2 Comments
Point 1: English language and style are fine/minor spell check required.
Response 1: The English language has been corrected.
Point 2: The paper presented an experimental study of a droplet impact on a sessile droplet previously placed on a hot wall. The temperature and velocity fields were measured by thermal imaging camera and PIV setups, respectively. The effect of surfactant on impact dynamics was studied as well. The manuscript has some novelty. The paper is well organized and written. I suggest publication after minor revisions.
Line 192 of pg 6, why did the authors choose to average over 20 maximum values? It seems that 20 is an arbitrary number.
Response 2: For reliable averaging of the velocity field, the averaging period should exceed the period of characteristic fluctuations hundreds to thousands of times (0.1 s*1000=100 s). In terms of the statistical analysis, the convection process in the drop is non-stationary and the analysis of the transition process after the drop fall lasts 2 s. It is obvious that with such a rapid change in the average velocity of the drop, it is impossible to provide averaging for hundreds of seconds. Therefore, an arbitrary averaging is of interest to show the difference between the maximum value and the "conventionally" average value. In addition, the following considerations are taken to select 20 averagings:
Umax(20) is the maximum velocity value resulting from the averaging of 20 maximum values of vectors (this number of vectors was chosen when processing the experimental results as the average under the condition of maintaining a uniform seeding of the velocity field; with a smaller number of vectors the error of determining the average velocity relative to those presented in this paper significantly increased, and a larger number was excessive for the experiments).
Point 3: How is constant kT (line 226 of pg 6) determined? If it is determined from authors' own data, it is no wonder they can get a good match of velocity predicted from the eq. 3 with experimental values.
Response 3: It is important to understand that the kT coefficient was taken not from this work (submitted for consideration), but from the author's previous works. These works are not related to the drop falling, but to the evaporation of a sessile drop on a hot wall, i.e. the experimental conditions were significantly different: different geometry of the drop, different wall temperatures, different qualitative temperature distribution on a free surface, etc. In addition, this coefficient was calculated based on the characteristic size. When the initial height of the drop is commensurate with the radius, then the height of the drop is taken as the characteristic size. This expression with kT can be effectively applied to layers when the drop height is ten times less than the layer diameter. Effective application of kT also takes place for films, which is presented in the articles published by the authors. However, in this case it is necessary to determine the statistical longitudinal integral scale. At that the kT value also remains. The results of other works are satisfactorily summarized using kT, which has been demonstrated in other articles by the authors.
Point 4: How is average temperature difference presented in fig. 4 calculated?
Response 4: The text has been added with the following paragraph:
“The average temperature was determined by ten different circles drawn inside the drop. This algorithm corresponded to the software. The circles were drawn at a certain distance from the contact line of the drop to exclude the influence of the wall. The average temperature was determined for each circle. The average surface temperature of the drop was calculated as the average value for 10 circles. The error of averaging using this method was less than the measuring error of the thermal imager.”
Point 5: I suggest the authors include images of dynamics process of droplet impact on sessile droplet, which can help better understand the effect of Weber number on velocity field measured by PIV.
Response 5: High-speed shooting of a drop falling is provided when the height of the falling drop is large enough, which leads to large Weber numbers, deforming the shape of the drop, splattering, corona formation, etc.
The real experiments did not aim to study the drop height and the Weber number. On the contrary, the purpose of the experiment was to exclude the role of dynamic fall, and to keep the role of the heat factor. The small drop was located very close to the free surface of the sessile drop. As a result, after the fall of a small drop, the shape of a large drop did not change, but there were only fluctuations of the free surface and their study was not part of the article’s goal. Therefore, high-speed shooting is pointless, since it is not informative for this case.

Reviewer 3 Report
The manuscript deals with the impact of a drop falling onto a second drop sitting on a hot surface. The experimental studies are performed with pure water as well as with drops of surfactant solutions. The target of the work is to find out the mass transfer rate in the drops. It is shown that the addition of surfactant reduces the velocity inside the sessile drop. The work is of relevance for heat transfer problems.
The flow pattern in the drops is monitored by particle image velocimetry at various experimental conditions.
The manuscript looks like the firth revision with changes marked in yellow. As I was not reviewer of the original version, it is difficult to judge if all questions have been answered properly. In the present state, however, the manuscript can be published as is.
Author Response
Response to Reviewer 3 Comments
Point 1: English language and style are fine/minor spell check required.
The manuscript deals with the impact of a drop falling onto a second drop sitting on a hot surface. The experimental studies are performed with pure water as well as with drops of surfactant solutions. The target of the work is to find out the mass transfer rate in the drops. It is shown that the addition of surfactant reduces the velocity inside the sessile drop. The work is of relevance for heat transfer problems.
The flow pattern in the drops is monitored by particle image velocimetry at various experimental conditions.
The manuscript looks like the firth revision with changes marked in yellow. As I was not reviewer of the original version, it is difficult to judge if all questions have been answered properly. In the present state, however, the manuscript can be published as is.
Response 1: The English language has been corrected.

Reviewer 4 Report
The overall impression of the paper is generally positive. In my opinion its greatest strength is related to the experimental part and its novelty. However, the quality of the presentation needs to be improved. There seems to be some confusion and the whole manuscript should be presented in a more ordered form. First, the "material" needs to be more clearly described, then "method" - the experimental procedure, data acquisition, measuring devices, errors, etc. should also be more clearly presented. A more ordered form would also be welcome in the results/discussion' part of the manuscript. I believe that the material presented in the paper is worth publishing and should be interesting to the readers, however improving the presentation quality will add value to the paper.
In my opinion the following elements of the manuscript need to be corrected/improved:
- The introductory part contains 41 references, however I would expect a more details analysis of the state-of-the-art closely related to the focus of the paper. In this part the Authors should discuss in more detail the results of other scientists' works and how they relate to the present study. Based on this discussion the conclusion about the novelty of the present paper should be formed. While in the paragraph starting in line 86 ("Thus, the literature analysis has shown.....") the conclusion about the manuscript novelty is presented, but not fully supported by the literature review presented earlier.
- Figure 1 should be placed close to its first reference in the text (line 122). The same applies to other figures.
- The description of the experimental procedure is mixed-up or even 'chaotic' as mentioned earlier. It is undoubtedly clear to the Authors, but - for the sake of the readers’ understanding and a possible repetition of the test results by other researchers - I suggest to clearly write in section 2: precisely what and how was measured, include more technical data of measuring devices and etc.
- Discussion of measurement errors would be welcome (apart from very limited data in line 163)
- In terms of the English language, the paper is generally well written, however there are some grammar mistakes (for example: lines 13-14: "The fall (...) are considered" - should be "IS considered", lines 158, 159 and other: "a drop 2" - should be without "a"; or other such as line 412: "velcoity" - should be "velocity"), so a proofreading by a professional translator might be welcome.
Author Response
Response to Reviewer 4 Comments
Point 1: The overall impression of the paper is generally positive. In my opinion its greatest strength is related to the experimental part and its novelty. However, the quality of the presentation needs to be improved. There seems to be some confusion and the whole manuscript should be presented in a more ordered form. First, the "material" needs to be more clearly described, then "method" - the experimental procedure, data acquisition, measuring devices, errors, etc. should also be more clearly presented. A more ordered form would also be welcome in the results/discussion' part of the manuscript. I believe that the material presented in the paper is worth publishing and should be interesting to the readers, however improving the presentation quality will add value to the paper.
In my opinion the following elements of the manuscript need to be corrected/improved:
The introductory part contains 41 references, however I would expect a more details analysis of the state-of-the-art closely related to the focus of the paper. In this part the Authors should discuss in more detail the results of other scientists' works and how they relate to the present study. Based on this discussion the conclusion about the novelty of the present paper should be formed. While in the paragraph starting in line 86 ("Thus, the literature analysis has shown.....") the conclusion about the manuscript novelty is presented, but not fully supported by the literature review presented earlier.
Response 1: The article has been re-structured. The measurement method is now described in more detail. Measurement errors have been added.
The Introduction has been added with the following phrases:
"An analysis of the literature has shown that previous research was aimed at studying the drop shape behavior after the drop fell on a solid wall. The fall of drops on the liquid layer led to the formation of corona and splashes. There are practically no experimental data on the effect of an instantaneous local temperature jump on the free surface of the drop on the convection inside the drop. The question remains how the interaction of droplets and the indicated short-term temperature inhomogeneity affect the intensity and duration of convection, as well as what characteristic convective structures occur inside the droplet."
In Abstract and Conclusion, the novelty and practical value are highlighted in color.
Point 2: Figure 1 should be placed close to its first reference in the text (line 122). The same applies to other figures.
Response 2: The figures are rearranged in accordance with the reviewer's instructions.
Point 3: The description of the experimental procedure is mixed-up or even 'chaotic' as mentioned earlier. It is undoubtedly clear to the Authors, but - for the sake of the readers’ understanding and a possible repetition of the test results by other researchers - I suggest to clearly write in section 2: precisely what and how was measured, include more technical data of measuring devices and etc.
Response 3: The structure of this paragraph has been changed. Description of experiments and characteristics of devices have been added.
Point 4: Discussion of measurement errors would be welcome (apart from very limited data in line 163)
Response 4: Data on measurement errors have been added to the article.
Point 5: In terms of the English language, the paper is generally well written, however there are some grammar mistakes (for example: lines 13-14: "The fall (...) are considered" - should be "IS considered", lines 158, 159 and other: "a drop 2" - should be without "a"; or other such as line 412: "velcoity" - should be "velocity"), so a proofreading by a professional translator might be welcome.
Response 5: The English language has been corrected.

This manuscript is a resubmission of an earlier submission. The following is a list of the peer review reports and author responses from that submission.
Round 1
Reviewer 1 Report
In this manuscript of Misyura et al., the authors investigated how the impingement of a small drop (Drop 1) affects a larger drop (Drop 2) resting on a heated substrate, by measuring the two dimensional flow velocity field on a plane using micro PIV and the surface temperature field of Drop 2 using thermal imaging. They also investigated the effect of adding surfactants to Drop 1, and conducted particle tracing on Drop 2 along with some theoretical modeling. The authors have published several papers in past two years on similar or related topics, but I reviewer reviewed the manuscript as if I were a reader of the manuscript who has not read the authors’ previous publications and who has basic knowledge and research experiences in thermofluids fields. Although the manuscript contains many contents, I have found that the manuscript is not suitable for publication for the following reasons. Overall, the authors made definitive arguments without providing any evidence, and mixed their conjecture and observation in confusing ways. Although my decision on this manuscript is “reject,” I have included some suggestions in the follow list to help the authors.
- Writing
The way the manuscript is written is not friendly to readers, and the manuscript has quite many grammatical errors and typos. Also, the manuscript lacks required details for readers to follow the authors’ rationale and lines of thought. Several symbols are not defined at all throughout the manuscript (for example, d_d and V_d in line 50, m in Line 281), some are used and then later defined (for example, U_01 in Line 258, which is defined in Line 291), and some are inconsistent (for example, R in Line 177 should be R_o2, and U_aver(max) and U_c(s)max; they seem to be the same). All these small mistakes and rough sentences make it hard to read the manuscript and to evaluate the scientific significance and rigor of the presented study. The authors are strongly recommended to consider general rules of scientific writing and graphic presentation.
Line 128: Drop 2 and drop 1 must be swapped.
Figure 1: It is better to include key dimensions such as D_o2, d_o2 and H and to show how the camera was positioned. Alpha α is shown in the figure, but never defined or referred to in the manuscript. The inset of PIV and TI images must be explained to help readers understand what these insets mean.
Line 129: What is this densimeter? Was this densimeter used to place drops? Lines 121-125 show that drops were dispensed using an electronic pipette.
Line 130: Surface tension is a quantity of an interface, so the surface tension coefficient must be defined for water-air.
Line 170-172: Three different speed quantities were used, but it is unclear what they represent in terms of physics and why all three speed quantities were used. In particular, the definition of U_max(20) is not straightforward. What does “averaging over 20 maximum values” mean? In one PIV result, did the authors chose 20 maximum speed values and average them? Why is such a quantity necessary?
It makes more sense to show PIV results first, and then to explain how three U quantities were calculated and to discuss U graphs.
Line 187-191: The authors suddenly introduce Ma_T and Ra, but they do not introduce the name of these dimensionless numbers and later they refer to these dimensionless numbers using their name.
Line 198-227: These paragraphs are incomprehensible. What is “the equation of motion” in Line 198? How can readers understand the following discussion or modeling, not know what this equation is? What does “linear approximation” in Line 200 mean? What is “nonlinear velocity term”? It seems that the authors had the same approach in their previous papers, but even if so, they have to provide readers with enough background and details so that readers can follow the authors’ argument. From the reviewer’s viewpoint, it is very hard to evaluate the soundness of the authors’ approach especially for these paragraphs, and it is very confusing whether they employed a very well established approach or not, since their arguments are so definitive and they cite their own papers.
Line 205: Although the authors argue that “the Rayleigh number is much less than the Ma number”, their numeric values are not shown.
Line 230-232: These sentences mention temporal change of T_s, but no data are given in the manuscript.
Line 252: What is “wall thickness”? Does it mean the thickness of the substrate for Drop 2?
Line 259-260: What does “the direction of rotation of Ma_T and Ra are the same”? Ma_T and Ra are dimensionless numbers. How do these scalar numbers have direction of rotation?
Figure 3(a): The arrows seem to indicate the direction of flow, but the figure is confusing because the arrows can be seen to indicate who Ma_T changes on the drop surface.
Line 285-286: Here, “pressure jump” is mentioned, but it is unclear whether the authors measured pressure (seems very challenging) in Drop 2 or just conjectured pressure change. If the latter is the case, it is better to reword the statements to show that these are conjectures, not what really happened. Many statements of the authors are so definitive that it is confusing whether their statement is about facts or conjecture.
Line 288-290: The authors mention “the law of conservation of momentum” and show a simple relation between rho*volume*speed. Did they use a control volume and the Reynolds transport theorem to obtain the equation? If so, they have to include details of their analysis, since the shown equation seems too simple (it looks like linear momentum conservation for a jet flow).
Line 338-346: It is unclear whether this part is based on the authors’ measurement/observation or theoretical estimation. For instance, the authors wrote “Drop 1 flows down from Drop 2 for about 0.01 s.” However, it seems quite hard to have this kind of observation because two drops are of water, so they will mix quite fast. Did the author have this observation using high speed imaging? If so, they can include images in supplementary information. Another example is that the authors employed boundary layer growth for their estimation, but they wrote their estimation in a very definitive tone, which gives an impression that they actually measured or saw the thermal boundary layer thickness.
Line 342: Although it is written that “l is shown in Figure 3(c)”, l is not shown in the figure.
Figure 5: A color bar of temperature should be included, and “drop 1” and “drop 2” must be swapped.
Some abbreviated words, especially surfactant names, must be spelled out for their first appearance.
Line 450, 462: Vorticity is mentioned here, but the authors do not provide any vorticity data. Vorticity does not always coincide with vortex, and if the authors wanted to use vorticity to describe their observation, they are supposed to show vorticity values measured from their PIV results.
Figure 11 caption needs to explain what 1, 2 and 1, 2, 3 are in (a) and (b).
Line 506: What is this “numerical simulation”? Does this mean solving Eq. (2) numerically?
Figure 12: Standard deviation values need to be shown in the graph, to show how repeatable 2-4 repeated experiments were.
Line 549: The authors argue that “the theoretical curve correctly reflects the experiment” in Figure 11(c), but in the following part, they admit noticeable difference between them for 0.3 < l/R < 0.8.
- PIV-related comments
As Figure 1 shows, the water-air interface of Drop 2 is not normal to the laser sheet, as shown by angle α, and this suggests that the curved interface must have refracted the laser sheet and thus the laser sheet could not be horizontal in Drop 2. How did the authors confirm that the laser sheet was horizontal in Drop 2? I have seen PIV studies to measure velocity profiles in drops, and in these studies, the laser sheet was vertical, passing the center of the drop. So, the laser sheet would not be refracted. Also, when PIV is employed for a flow in a curved pipe, the pipe is immersed in a larger container filled with the same working liquid, to prevent any refraction. In the current manuscript, it is very obvious that the drop was three dimensional and curved, and the water-air interface refracts light. Isn’t this refraction effect related with the non-symmetric shape of the drop cross-section shown in PIV result figures?
The used laser has a high energy level (70 mJ), and so irradiating the laser sheet onto Drop 2 may have caused changes in the surface tension. The PIV laser in my lab has a similar level of laser energy, and with the full energy (without using any ND filters), the laser sheet can burn dusts in air. Since the manuscript focuses on flow induced by spatial changes of surface tension, this high energy laser sheet may have affected surface-tension-driven flow. I wonder if the authors have checked any effect of laser power on their study.
Also, what was the thickness of the laser sheet? According to this manuscript, the height of Drop 2 was about 2 mm, and the distance from the substrate wall to the laser sheet was about 0.2 mm. This implies that the laser sheet thickness should be in the order of 0.1 mm, which seems quite challenging.
The PIV section in page 4 does not have any information about seeding particle (such as material, density, particle size, number density).
Figure 4: (1) The shown horizontal cross-section of Drop 2 is not circular. Why? Since Drop 2 was resting on the substrate of the room temperature, the horizontal cross-section must be circular. (2) In the top row, who do speed maps show several local spots of higher speed (around 0.3 m/s, while the rest is about 0 m/s). Is this an error in PIV measurement or post-processing? (3) In Line 324, the authors wrote “there are 5-6 vortices” in Figure 4, but such number of small vortices cannot be seen in the figure. The authors are encouraged to use arrows to indicate these vortices.
- Other experiment-related comments
The authors mentioned that they used high speed imaging a few times in the manuscript, but no detailed information on imaging condition (such as frame rate, magnification ratio, and so on) is provided, which makes it hard for readers to judge the quality of imaging.
Line 138: How was the static contact angle measured?
This manuscript focuses on surface-tension-driven flow due to temperature gradient. However, the authors do not provide any data about surface tension coefficient. How can reviewers and readers evaluate the authors’ data? For example, calculating the Marangoni number defined in Line 187 requires the temperature gradient of surface tension coefficient, but the number is not shown. Also, Line 452 discusses spatial surface tension gradient, but no value is offered. Last, the authors tested the effect of three surfactants in their study and discussed the effect of lowered surface tension (for instance, Line 420). However, they do not provide any numeric values of decreased surface tension coefficient due to surfactants.
In the same sense, the fluid property values for the Rayleigh number are not provided. In addition, it is unclear how the authors considered the temperature dependence of these properties. For example, the viscosity of water changes with temperature, and in this manuscript, Drop 2 experienced significant temperature changes and temperature gradient.
Line 292-293: The authors mention “two symmetric and oppositely rotating vortices” in the vertical section, presumably, of Drop 2. If this is what the authors observed, how did they observe these vortices? The authors measured velocity field on the horizontal cross-section, not vertical section. Also, simply dividing “the pulse of the falling drop” with 2 does not make sense, because the studies drop system was axisymmetric and thus the impact from Drop 1 spreads in the radial direction and possible vertical structure is toroidal.
Figure 5 shows that the smaller drop was offset from the larger drop, which implies that the thermal imaging camera was looking at the drops with an angle, not from the direct top. This configuration is ok with temp measurement, but it raises a question whether the PIV camera was also used in the same configuration. This is why Figure 1 should include these cameras to clearly show how imaging was done.
Line 395-398: All surfactants were added to Drop 1 above the critical concentration, so micelles are expected to have existed in Drop 1. The authors appear to have chosen such concentrations so that Drop 1 was fully saturated with surfactant molecules. However, once Drop 1 merged with Drop 2, the concentration of surfactant molecules must have decreased significantly, which is due to large volume difference between the drops, and thus the surfactant concentration must have dropped below the critical concentration. The authors are expected to consider this effect or discuss any effect of sudden change in the surfactant concentration in their experiment.
Figure 7: (1) In Line 428-429, the authors wrote that a toroidal vortex existing, which is concentric with the drop, only in the -0.5 s case. However, it can be easily expected that such toroidal vortex would exist in Drop 2 steadily before the impingement of Drop 1, but other images in the top row do not show such idea vortex. Also, the top rows of Figures 8-10 do not show this ideal toroidal vortex. Was Drop 2 in the unsteady state by nature in this experiment? Why? Maybe due to evaporation? If so, did this unsteadiness affect PIV measurement or repeatability of the study? (2) Line 431 mentions “A single main vortex” seen in the 0.75 s image, and the rotational axis of this vortex seems to be horizontal (so, the image shows velocity vectors all in one direction). This can cause a confusion to readers, so the authors are recommended to describe vortices seen in their PIV result more clearly and distinctively. (3) Line 434 mentions “5-7 vortices” in the 0.25 s image, but the image does not show that many vortices. The authors are encouraged to indicate these vortices in the image to help readers.
Metal particle tracing and the force balance model [Eq. (2)]: (1) More information (such metal type and density) of the used metal particles should be included. (2) Eq. (2) should include gravitational forces (buoyancy and weight) of the particle, because these forces have components parallel with F_st. (3) As the particle moved upward along the free surface of the drop, it experienced varying temperature, which means that surface tension on the particle also changed along its trajectory. How was such effect considered in this modeling? (4) Solving Eq. (2) requires how V_l changes along the drop surface. But no information or assumption is given. How did the authors treat V_l in Eq. (2)?
Reviewer 2 Report
This paper presents very interesting research into identifying the role of the inertia forces thermocapillary forces for the generation of convection in the liquid when a drop falls on a larger sessile drop located on a hot wall. Moreover, the effect of surfactants on heat and mass transfer in a drop is investigated. The Micro-Particle Image Velocimetry, as well as thermal imager, were applied during the experimental research. The strength of the paper is the soundness of methodology as well as high scientific and applicable value. The paper is well organized and easy to read. The introduction section covers the necessary technical background and refers to up-to-date literature but the main novelty of the paper should be better emphasized in this section. The research object is properly described. However, more detailed information concerning the selection of the horizontal (only) section of the sessile drop 0.15-0.2 mm from the substrate surface would be beneficial. The results are thoroughly discussed but authors should elaborate on why they refer to maximum velocity and at the same time they conclude that it is incorrect to perform calculations based on maximum velocity (line 243-246). The sentence in lines 450-452 is unclear. In my opinion “uniform” should be replaced with “ununiform”. The presented conclusions are well justified and properly drown based on the research results. I also recommend emphasizing the novelty of the research in the conclusions.
Reviewer 3 Report
The manuscript deals with important heat and mass transport phenomena in droplet (coalescence) including also surfactants. That said, many different factors make this problem the same time more interesting and also more complicated. Although my background is in theory and simulation, I appreciate that the experimental part being familiar with the experimental methods. The manuscript is well-written and results are sound and clear on a very important and novel topic. Methods are appropriate. Figures and references are also appropriate. Overall, it is a very nice paper. I have no reservations to recommend publication of this article in Applied Sciences. An experimental referee could get better down to the details of the experimental methodology though.